# Three-dimensional scanless holographic optogenetics with temporal focusing (3D-SHOT)

Nicolas C. Pégard [1,2], Alan R. Mardinly [1], Ian Antón Oldenburg [1], Savitha Sridharan[1], Laura Waller [2] & Hillel Adesnik [1,3]

Optical methods capable of manipulating neural activity with cellular resolution and millisecond precision in three dimensions will accelerate the pace of neuroscience research. Existing approaches for targeting individual neurons, however, fall short of these requirements. Here we present a new multiphoton photo-excitation method, termed three-dimensional scanless holographic optogenetics with temporal focusing (3D-SHOT), which allows precise, simultaneous photo-activation of arbitrary sets of neurons anywhere within the addressable volume of a microscope. This technique uses point-cloud holography to place multiple copies of a temporally focused disc matching the dimensions of a neuron's cell body. Experiments in cultured cells, brain slices, and in living mice demonstrate single-neuron spatial resolution even when optically targeting randomly distributed groups of neurons in 3D. This approach opens new avenues for mapping and manipulating neural circuits, allowing a real-time, cellular resolution interface to the brain.

[1] Department of Molecular and Cell Biology, 205 Life Science Addition, University of California, Berkeley, CA 94720, USA. [2] Department of Electrical Engineering and Computer Science, 514 Cory Hall, University of California, Berkeley, CA 94720, USA. [3] Helen Wills Neuroscience Institute, 132 Barker Hall #3190, University of California, Berkeley, CA 94720, USA. Nicolas C. Pégard and Alan R. Mardinly contributed equally to this work. Correspondence and requests for materials should be addressed to H.A. (email: hadesnik@berkeley.edu)

Optical manipulation of neural circuits is one of the most powerful approaches for revealing causal links between neural activity and behavior[1, 2]. Optogenetics enables rapid and reversible control of genetically defined cell types by employing photo-sensitive microbial opsins that either generate or suppress neuronal activity in response to light[2–4]. In principle, optogenetics offers high spatiotemporal precision, yet the vast majority of optogenetics studies primarily leverage genetic specificity rather than high resolution spatial control due to the difficulties of accurately focusing light in brain tissue. However, since many neural computations and behaviors rely on populations of neurons that are genetically similar but spatially intermixed[5–8], new methods are needed to enable precise three-dimensional (3D) targeting of custom neuron ensembles within the brain.

Several methods have been developed for optogenetic photostimulation; however none are capable of simultaneous single-neuron spatial resolution[9, 10] across a large volume without compromising temporal precision. The simplest approach, one-photon optogenetics, uses absorption in the visible spectrum to activate the opsin, yet strong optical aberrations and scattering through brain tissue severely degrade spatial resolution, even when using adaptive optics[11]. Two-photon (2P) excitation partially addresses the issue of optical scattering[12], dramatically improving axial resolution and depth penetration of the illumination patterns[13, 14]. The most common approach employed for two-photon excitation is to focus a femtosecond-pulsed infrared laser beam into a single diffraction-limited spot which is scanned in two-dimensions (2D) or three dimensions (3D)[14, 15]. For optogenetic applications, a raster[16, 17] or spiral[5, 18, 19] pattern is scanned across the cell body of each targeted opsin-expressing neuron. Since two-photon absorption is nonlinear, photoexcitation is confined to a small spot, enabling single neuron spatial resolution at appropriate power levels[16, 18]. However, photosensitive opsins such as ChR2 deactivate rapidly, making it difficult to quickly achieve the photocurrent needed for fast and reliable action potential initiation by activating the opsin point by point in each neuron[18]. While newly engineered opsins with slow deactivation kinetics overcome this problem[17], they necessarily degrade temporal resolution, making it difficult to trigger action potentials with precise timing[16, 19].

Computer generated holography (CGH)[20–23] is a scanless solution for two-photon optogenetics[9, 24–29], which significantly increases temporal precision of photostimulation. CGH relies on a spatial light modulator (SLM) to distribute a laser beam into multiple targets with custom 3D shapes[20, 21]; unlike scanning approaches, CGH wide-area holograms matched to the dimensions of each neuron's soma should enable simultaneous, flash-based activation of large numbers of opsin molecules yielding photocurrents with fast kinetics[10]. However with CGH, spatial resolution along the optical axis is entirely determined by the rate by which wave propagation attenuates the power density on either side of the targeted area. Thus, CGH favors high numerical aperture (NA) objectives with small addressable volumes. CGH, and even point scanning methods[10, 18], often result in significant undesired photoexcitation above and below the object target. In practice, physiological spatial resolution is highly power dependent, and single neuron spatial resolution (a Gaussian fit axial full-width at half-maximum (FWHM) of ~30 μm[9, 10]) is generally impossible across large volumes.

Temporal Focusing (TF)[30] eliminates the trade-off between the target size in the lateral (*x,y*) plane and axial (*z*) resolution[30, 31]. TF relies on a diffraction grating placed in the image plane to decompose femtosecond pulses into separate colors, such that the different wavelengths components within the original pulse propagate along separate light paths. Each component of the

decomposed pulse is broadened in time, which dramatically reduces peak intensity and prevents two-photon absorption until the original pulse constructively interferes at the conjugate image plane of the diffraction grating[30, 31]. This strategy introduces a secondary axial confinement mechanism that restricts the two-photon response to a thin layer around the focal plane. Since the thickness of this layer no longer depends on the dimensions of the targeted area, but rather on the bandwidth of the pulse, this property has been successfully applied for depth-selective two-photon fluorescence imaging[32, 33], and has been implemented with mechanical scanning[34] and with random-access volume sampling of functional fluorescence[35].

For two-photon photostimulation applications, TF activates opsins over a wide area matching the neuron's shape in the focal plane, without compromising depth specificity[10, 36]. TF also mitigates the effects of light scattering[37, 38] even through thick layers of brain tissue[39, 40]. Although multiphoton CGH with TF can achieve wide field photostimulation with cellular resolution and high temporal precision, most implementations only enable excitation within a single 2D plane[10, 36, 41]. Thus, neurons located above or below the focal plane are not addressable, a necessary condition for many experiments designed to interface with neural circuits in vivo, where neurons are located continuously in 3D. Multi-level temporal focusing has been shown with holograms tiled into clusters that can be individually defocused in space by applying digital lens patterns on a second SLM. However, this strategy is limited to at most 4–5 depth levels before in-plane resolution is degraded, severely constraining the neuronal population that is simultaneously addressable with optical stimuli[42]. In most brain structures, neurons are distributed in 3D, not in discrete layers. Therefore, the inability of scanless optogenetics approaches to target neurons at any arbitrary set of axial planes simultaneously is a major obstacle for large-scale optogenetic interrogation of neural circuits.

To overcome this outstanding challenge, we developed 3D scanless holographic optogenetics with temporal focusing (3D-SHOT), a new holographic photostimulation technique that combines CGH and temporal focusing to enable single-shot in vivo photo-activation of custom neuron ensembles with single neuron spatial resolution, without limits on the number of addressable target planes. Our strategy benefits from the large accessible volume of conventional holographic microscopy, yet takes advantage of temporal focusing to further confine the two-photon photoexcitation in the axial dimension. Here we present the theoretical design and empirically validate the ability of 3D-SHOT to target neurons in an arbitrary number of axial planes with high spatial precision. We further demonstrate its functionality for in vivo optogenetics applications in mice, validating 3D-SHOT's utility as a new technology for scanless optical manipulation of neural activity in the intact brain.

## Results

**Designing a TF system for arbitrary 3D targeting**. 3D-SHOT operates by replicating identical copies of a temporally focused disc of light, termed the custom temporarily focused pattern (CTFP), at arbitrary 3D positions. Here, the dimensions of the CTFP were chosen to match the characteristic size of a layer 2/3 cortical pyramidal neuron[35], (Supplementary Table 1) and may be easily adjusted for different applications (Supplementary Table 2). The CTFP is designed by separating a patterned laser beam (characterized by its phase and intensity) into spectral components with a diffraction grating as in a conventional TF system. The beam intensity profile determines the dimensions of the CTFP, while phase, a previously unused degree of freedom, is engineered to make 3D holography and temporal focusing

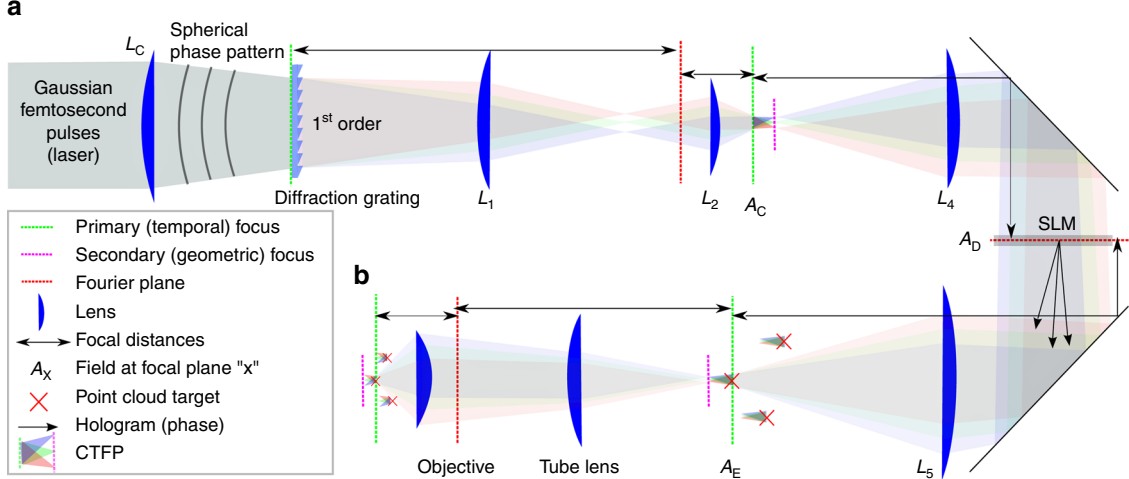

**Fig. 1** Simplified experimental set-up for 3D-SHOT. **a** A spherical phase mask is applied to a Gaussian femtosecond laser pulse incident on a blazed diffraction grating. After the grating, the direction of propagation of the first-order beam is wavelength-specific, decomposing the pulse in the temporal domain. Temporal focusing and the associated nonlinear response thus only happens across a disc-shaped area at depth planes corresponding to images of the diffraction grating (dashed, green). We call this 3D intensity distribution a Custom Temporally Focused Pattern (CTFP). **b** We then copied this temporally focused disc pattern to different areas in 3D by computer generated holography (CGH). To do so, a spatial light modulator (SLM) in Fourier space (dashed, red) generates a point-cloud hologram that replicates the CTFP at each target point in the volume image. The resulting field is demagnified again before impinging on the sample to create custom 3D positioning of the temporally focused disk, each targeted at a particular neuron

compatible. To ensure good diffraction efficiency of all spectral components by the SLM, we used lens $L_c$ to apply a small spherical phase pattern. The focal length was adjusted so that each spectral component of the pulse spans across the short axis of the SLM in the Fourier domain (Supplementary Figs. 1, 2). Besides experimental simplicity, the spherical phase mask imposed by lens $L_c$ yields convenient analytical expressions for simulation purposes (Supplementary Notes 1–3). We note, however, that alternate phase masks (for instance generated with a secondary SLM), could also be employed as long as they expand the spectral components to cover enough area of the SLM to enable holography.

The fundamental principle of 3D-SHOT (see Methods section) is to make temporal focusing compatible with 3D CGH by placing the SLM after the diffraction grating. This design forgoes the ability to synthesize custom shapes for each neuron, but represents an acceptable trade-off for many optogenetic applications. The resulting experimental set-up operates an all-optical convolution product given by:

$$A_E(x, y, z, \Delta k) = A_C\left(\frac{f_4 x}{f_5}, \frac{f_4 y}{f_5}, \Delta k\right) \otimes \left[F\left[e^{i\varphi}\right]\left(\frac{x}{\lambda f_5}, \frac{y}{\lambda f_5}\right) \otimes h(x, y, z)\right],$$
(1)

where $\otimes$ denotes the convolution product. The CTFP, $A_C(x, y, \Delta k)$, adjusted to the characteristic dimensions of a neuron after demagnification, can then be placed on demand at the desired locations by an all-optical convolution with a 3D hologram corresponding to the phase mask, $\varphi$, displayed on the SLM (Fig. 1). To simultaneously place identical copies of the CTFP at $n$ neurons located in positions $(x_i, y_i, z_i)$ in space, we computed a hologram optimized so that the Fresnel propagation, $h$, of $F[e^{i\varphi}]$ best approximates a cloud of points located at the desired targets with weighted intensities that can be adjusted for each targeted neuron. A demagnified copy of the CTFP was then recreated at each point of the cloud in the resulting field.

Here, our primary goal was not to render a visually accurate hologram but instead to increase contrast for two-photon excitation at selected locations while avoiding inadvertent photo-activation of non-targeted areas, and we implemented a modified version of the multilevel Gerchberg–Saxton (GS)[20] algorithm that emphasizes precise control of the intensity at the desired targets.

**Quantitative 3D characterization of two-photon absorption.** To evaluate the capabilities of 3D-SHOT and quantify how two-photon absorption is spatially distributed in 3D, we placed a flat phase mask on the SLM, $\varphi(x, y) = 0$, so that the convolution product in Equation (1) reduces to the simplest case of requesting a single copy of the CTFP to be placed at the center of the operating volume, defined in $x = y = z = 0$ (that is, the system's zero-order). We then placed a thin fluorescent film on a microscope slide under the excitation objective and recorded the corresponding two-photon fluorescence with a sub-stage objective coupled to a camera (Fig. 2a). We recorded $z$-stack images by moving the excitation objective along the $z$-axis by 1 μm increments. The resulting data correspond to a quantitative 3D measurement of two-photon absorption induced by the CTFP. In the specific case of a flat phase pattern on the SLM, we have also derived an analytical expression for the instantaneous two-photon absorption, $|A_F(x, y, z, \Delta t)|^4$ (Supplementary Fig. 3), which we used to predict and visualize the CTFP in space and time, and also to compare simulation and experiments.

We first compared 3D-SHOT to conventional 3D holography (Fig. 2b). Using CGH, we computed a 10 μm disk image target at $z = 0$ where we imposed a high-frequency speckle pattern to maximize spatial confinement along the $z$-axis. Projection views of two-photon absorption along the $x$, $y$, and $z$ axes show how even in an optimized hologram unwanted photostimulation remains above and below a neuron targeted with this method. When the 10 μm CGH pattern is replaced by a 3D-SHOT CTFP of equal size, experimental results (Fig. 2c) and simulations (Fig. 2d–e) show that temporal focusing significantly enhances spatial confinement along the $z$-axis.

**Scanless two-photon optogenetics using 3D-SHOT.** We evaluated the spatial resolution of 3D-SHOT via optogenetic stimulation of opsin-expressing Chinese Hamster Ovary (CHO) cells and mouse neurons in acute brain slices and in vivo. We

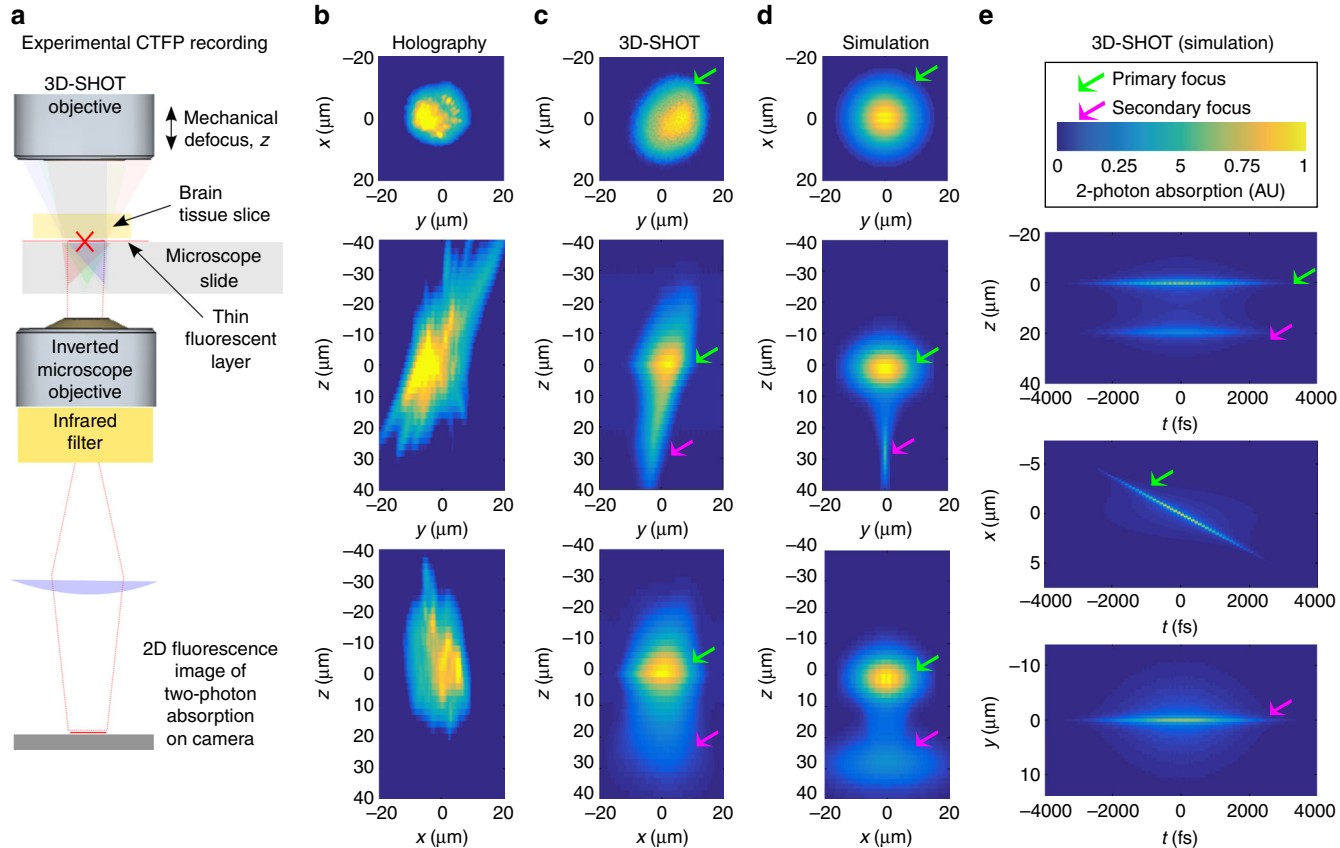

**Fig. 2** Optical characterization of the spatial resolution of CGH vs. 3D-SHOT. **a** We used a fluorescent calibration slide and an inverted microscope to compare two-photon absorption patterns in 3D. **b** For conventional holography, we consider a 10 μm diameter disk target, and show (from top to bottom) example projection views of two-photon absorption in the $(x,y)$, $(z,y)$, and $(z,x)$ planes. With 3D-SHOT, the CTFP was adjusted to a 10 μm diameter target and the same projection views were recorded **c** experimentally, and **d** rendered with simulation. **e** Simulation additionally provides space-time projections (Supplementary Notes 1–3). The primary focus (green arrow) at $z = 0$ displays the characteristic depth-sectioned properties of temporal focusing, with a perceived femtosecond pulse line object rapidly swept along the $x$-axis. The secondary focus (pink arrow), a static line along the $x$-axis in $y = 0$, is a geometric projection of the spherical phase pattern induced by lens $L_c$. The pulse duration at the secondary focus is stretched in time (~1000 fs) which minimizes nonlinear response

evaluated the spatial resolution (or physiological point spread function (PPSF)) by recording the photocurrent response to multiphoton photostimulation as a function of the displacement between the holographic target and the patched cell (Fig. 3a, e). The set-up is fitted with a pair of mirrors on a sliding stage to rapidly swap between CGH and 3D-SHOT (Supplementary Fig. 1). The two optical paths are aligned so that the centers of the CTFP (3D-SHOT path) and the disc illumination (CGH path) are co-aligned along all 3D. This allows both methods to be tested on individual cells without any realignment.

Since power levels needed for photostimulation vary across cells due to differences in opsin expression and excitability, we compared the spatial confinement of 3D-SHOT and CGH photo-excitation as a function of laser power density. With conventional holography, we observed substantial photocurrents 25–50 μm above and below the target, indicating that photo-activation of non-targeted neurons is likely to occur (Fig. 3b, c). As predicted by simulations (Supplementary Fig. 3), temporal pulse stretching significantly enhanced spatial resolution with 3D-SHOT, as photocurrents were more significantly attenuated above and below the primary focus (Fig. 3f, g). In neurons and CHO cells, axial resolution with 3D-SHOT was significantly improved relative to CGH, even when using several orders of magnitude more laser power (Fig. 3d, h). This implies that one may increase excitation light to generate action potentials without compromising spatial confinement.

**3D-SHOT photostimulation with single-neuron resolution.** We next quantified the physiological spatial resolution of CGH and 3D-SHOT in neurons by measuring the spiking probability along the lateral $(x,y)$ and axial $(z)$ dimensions. Experimental results show similar spatial resolution with both methods in the lateral direction in the focal plane, with a FWHM of $10 \pm 2$ μm for holography, and $9$ μm $\pm 1.3$ for 3D-SHOT, ($p = 0.57$, Mann–Whitney $U$-test) consistent with the dimensions of the disc and CTFP (Fig. 4a). With CGH, the spike probability along the $z$-axis does not permit single-cell resolution (FWHM $= 78 \pm 6$ μm). In contrast, 3D-SHOT provides far superior resolution (FWHM $= 28 \pm 0.7$ μm, $p = 0.006$, Mann–Whitney $U$-test, Fig. 4b, c) compatible with single-cell resolution in 3D, in that the FWHM of spike probability is on par with the typical dimensions of a cortical neuron and their inter-somatic spacing. 3D-SHOT also has a secondary focus that is sheared off-axis relative to the linear $(x,y,z)$ dimensions (Fig. 2). Although, we did not predict significant two-photon excitation to occur at the secondary geometric focus, we tested the spatial resolution by photo-stimulating a 3D grid pattern ($50 \times 50 \times 100$ μm³ $x,y,z$). This experiment revealed that the neuron was photo-activated only when the disc image was targeted to the cell body (Fig. 4d).

To demonstrate 3D-SHOT's single-neuron spatial resolution under in vivo conditions, we quantified the PPSF for L2/3 pyramidal neurons using 3D-SHOT and CGH in living mice. We obtained two-photon guided loose patch recordings from opsin-

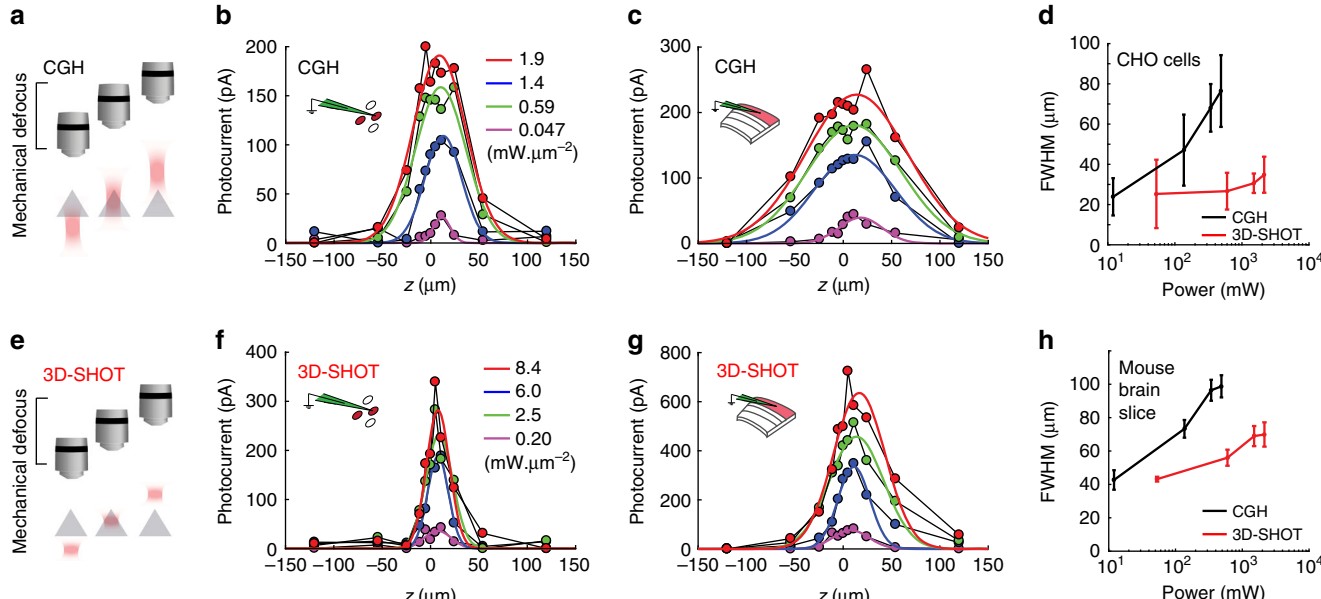

**Fig. 3** 3D-SHOT generates axially confined photo-activation. **a** A photostimulation pattern generated with CGH was mechanically stepped along the optical $z$-axis and passed through a cell expressing opsin. Photocurrents were recorded in the whole-cell voltage clamp configuration. **b**–**f** The response profile for CGH with a 10 μm disk target and different power levels on CHO cells for CGH **b** or 3D-SHOT **f** ($n = 12$ cells, data points are a representative example cell). **d** The FWHM of the characteristic response profile for both methods at various power levels on CHO cells ($n = 12$ cells, data represent mean ± s.e.m.). **e** As in **a**, but in mouse brain slices. **c**, **g** as in **b**, **f**, but in mouse brain slices ($n = 11$ neuron, data points are a representative example neuron). **h** The FWHM of the characteristic response profile for both methods at various power levels in mouse brain slices ($n = 11$ neurons, data represent mean ± s.e.m.)

expressing neurons in anesthetized mice, generated action potentials in targeted neurons with light pulses, and measured the PPSF for CGH and 3D-SHOT by digitally displacing the holographic target. As in brain slices, CGH and 3D-SHOT exhibited similar spatial resolution in the lateral ($x$,$y$) dimensions (lateral FWHM: CGH: 15 ± 4 μm, 3D-SHOT: 11 ± 2 μm, $p = 0.46$, Mann–Whitney $U$-test, Fig. 4e). However, along the $z$-axis, 3D-SHOT achieved significantly better spatial confinement than CGH (Axial FWHM: CGH: 70 ± 16 μm, 3D-SHOT: 29 ± 3 μm, $p = 0.004$, Mann–Whitney $U$-test, Fig. 4f, g). Measuring spike probability in response to 3D-SHOT stimulation throughout a 3D $50 \times 50 \times 100\ \mu m^3$ grid revealed no off-axis excitation by the secondary focus (Fig. 4h). Together, these data validate 3D-SHOT as a novel scanless optogenetic stimulation paradigm that can achieve single-neuron spatial resolution during optogenetic stimulation in the mouse brain in vivo.

**Spatially precise remote control with 3D-SHOT.** Since the major advantage of 3D-SHOT is the ability to target multiple neurons arbitrarily in 3D, it is vital that 3D-SHOT can activate neurons with high spatial resolution even when digitally focusing light far from the zero-order of the optical system. Therefore we next evaluated the accessible depth within the volume by measuring the activation and spatial resolution as a function of the distance from the holographic natural focus plane. Toward this end, we recorded photocurrents in CHO cells via a voltage clamp and then measured spike probability in neurons via a current clamp in mouse brain slices (Fig. 5a, e). To test whether the CTFP can be digitally displaced along the $z$-axis, we systematically moved the digital focus of the hologram, and accordingly corrected the mechanical position of the objective by the same distance ($\delta z_{\text{Digital}} = -\delta z_{\text{Mechanical}}$). This test showed that 3D-SHOT effectively photostimulates cells at locations distal to the zero-order, as photocurrent and spike-probability were not affected by digital offset in $z$ (Fig. 5b, CHO cells $p = 0.39$, Fig. 5f, neurons $p = 0.2$, Kruskal–Wallis).

We next asked if the axial resolution of stimulation was constant when stimulating away from the natural focal plane at $z = 0$. For this we measured the FWHM of the axial PPSF as a function of digital defocus. As before, we digitally moved the holographic target along the $z$-axis, but instead of matching the digital and mechanical offset, we stepped the objective across the entire range of the $z$-axis and measured the physiological response at each location. This allowed us to measure the axial resolution of photostimulation from locations distributed on either side of the optical axis. Results show that 3D-SHOT effectively confines excitation to the desired depth range throughout the 180 μm range that we sampled, as the FWHM of stimulation did not change as a function of digital defocus in $z$ (Fig. 5c-d, CHO cells $p = 0.07$, neurons, Fig. 5g-h. $p = 0.17$, Kruskal–Wallis test). These experiments show that 3D-SHOT retains axial confinement capabilities compatible with single-cell resolution for photocurrents and spike probability while targeting neurons at any depth within the accessible volume defined by the SLM and the microscope assembly (Supplementary Fig. 4, Supplementary Note 4).

**Selective photostimulation of vertically stacked neurons.** We next considered the particularly challenging scenario of activating two vertically stacked neurons without activating a neuron located between them. To test this scenario, we computed holograms that simultaneously target two axially aligned targets vertically separated by about 80 μm along the optical $z$-axis, either two disc patterns with conventional 3D CGH or two copies of the CTFP target made with 3D-SHOT and a two-point hologram. We then compared the photo-induced response in a single patched cell (CHO cells and neurons). Since the typical size of a L2/3 cortical pyramidal neuron is 15–20 μm in the axial dimension (plus additional strong photocurrent contributions from an apical dendrite oriented along the optical axis)[43], without an ($x$,$y$) offset we expect that two pyramidal neurons would rarely be packed closer than 80 μm in $z$ with a third neuron positioned between

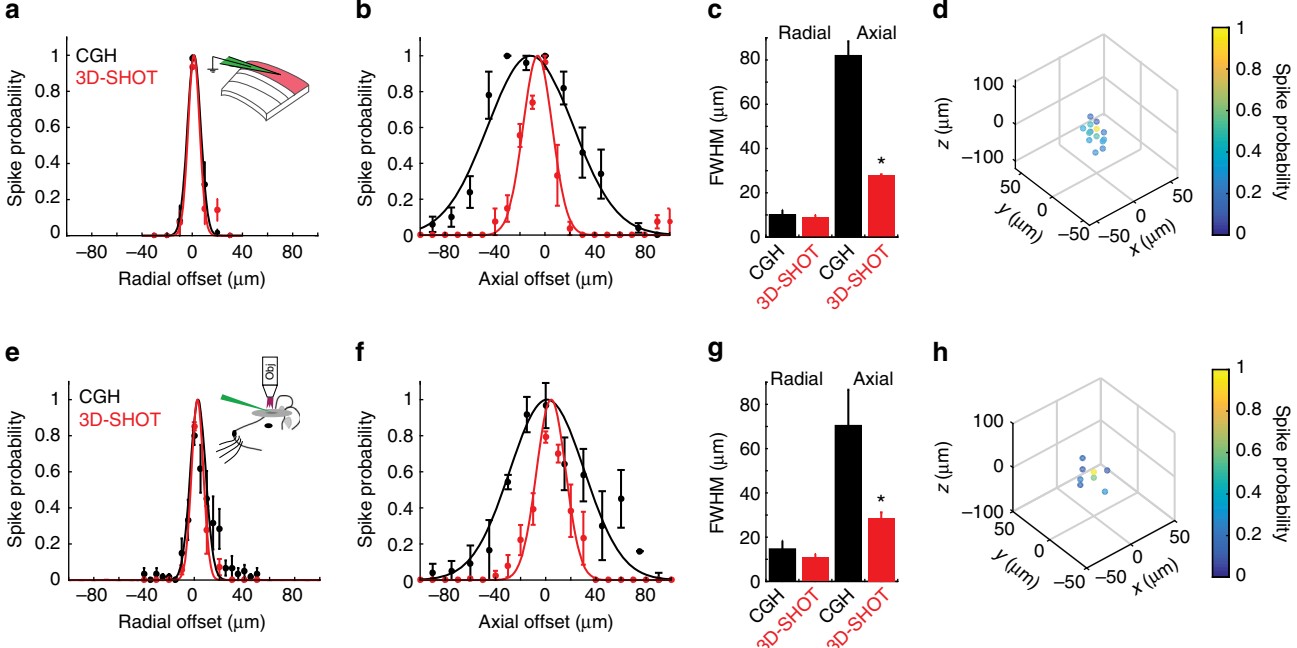

**Fig. 4** 3D-SHOT provides high spatial resolution photo-activation of neurons in vitro and in vivo. **a** Spatial profile of two-photon evoked spiking of a L2/3 pyramidal neuron in a mouse brain slice in the radial dimension. (CGH: black, $n = 12$ neurons; 3D-SHOT: red, $n = 14$ neurons; $p = 0.56$ Mann−Whitney $U$-test. Data represent mean ± s.e.m). **b** As in **a**) but along the axial dimension (CGH $n = 11$ neurons, 3D-SHOT $n = 5$ neurons, $p = 0.006$). **c** Quantification of the FWHM comparing CGH and 3D-SHOT (data represent mean and s.e.m.). **d** Full volumetric assessment of photostimulation resolution using 3D-SHOT. Points throughout the volume were tested, but only points that elicited spike probability greater than zero are shown (data represent mean of $n = 7$ neurons). **e−g** As in **a−c** but for neurons recorded via in vivo 2P guided patch: radial: $p = 0.46$; axial: $p = 0.004$, CGH ($n = 6$ neurons), and 3D-SHOT ($n = 6$ neurons). **h** as in **d**, but in vivo cell-attached patch using 3D-SHOT (data represent mean of $n = 5$ neurons)

them[44–46]. Thus this scenario represents a challenging, yet physiologically relevant test case.

As in the previous experiment, a CHO cell or neuron was patched to record photocurrent and/or spike probability as a function of the respective displacement between the cell and the photostimulation pattern with two targets on the optical axis (Fig. 6a). With conventional holography and two disk targets, we observed a significant amount of photocurrent when the cell is located between the two targets (Fig. 6b, c), where no photocurrent is desired. Conversely, with 3D-SHOT and two similarly axially separated copies of the CTFP, no photocurrent was observed at the intermediate location, and thus spatial resolution was significantly improved (Fig. 6b, c). In neurons, the added non-linearity introduced by the cell's action potential threshold further enhanced the axial contrast in terms of spike probability (Fig. 6d, Supplementary Fig. 5, Supplementary Note 5).

**Addressing arbitrary 3D locations with precise power control.** Our results so far establish that 3D-SHOT allows optogenetic activation of neurons with high spatial precision in vivo and can arbitrarily target light in 3D with high spatial resolution. However the major advantage of 3D-SHOT over existing approaches is its ability to simultaneously illuminate many regions of interest located throughout a volume, each at a different axial $z$-depth. To demonstrate the ability of 3D-SHOT to simultaneously target disc images to 50 unique $z$-planes, we computed a phase-mask corresponding to a point-cloud hologram with 50 targets placed in a spiral pattern, each with a unique $z$-position. We measured the two-photon absorption in the vicinity of each targeted spot and in 3D with a sub-stage camera (as in Fig. 2, but repeatedly near each

target), demonstrating that 3D-SHOT can illuminate this complex set of 50 targets at 50 separate axial depths (Fig. 7a).

However, we note that while each spot in this 50 region of interest hologram clearly displays a replica image of the CTFP at each targeted point, the total two-photon energy deposited from target to target exhibits a high variance (27%). Precise power control on each target represents a critical challenge for 3D-SHOT, since both the neuronal response and the spatial resolution are related to the stimulation power (Fig. 3). In holographic optical systems such as 3D-SHOT, spatial non-uniformities in the distribution of power are related not only to well-known photon losses throughout the optical system, but also to the SLM's diffraction efficiency and its ability to accurately render specific CGH patterns.

For precise control of the power distribution across multiple targets, we have developed a model of the diffraction efficiency as a function of target location, and we relied on simulations during hologram computation to compensate for the known spatial dependence of the SLM's diffraction efficiency. As for additional physical losses through the optical system, we measured the difference between expected and actual received power at 15,000 discrete points randomly distributed throughout our addressable volume (Supplementary Fig. 6). We then generated a 3D interpolant to adjust the weights for each spot in multi-target holograms (Fig. 7b). By combining simulation and experimental calibration data, our system compensates for non-uniform power distributions (Supplementary Note 6). To test the performance of this approach, we re-computed a phase mask targeting the same 50 spots with power compensation and measured 2P absorption as before. The results show a dramatic reduction in the variance (down to 9%) of the 2P absorption in each spot (Fig. 7c−e, $p < 1 \times 10^{-12}$ $F$-test), validating that 3D-SHOT used in

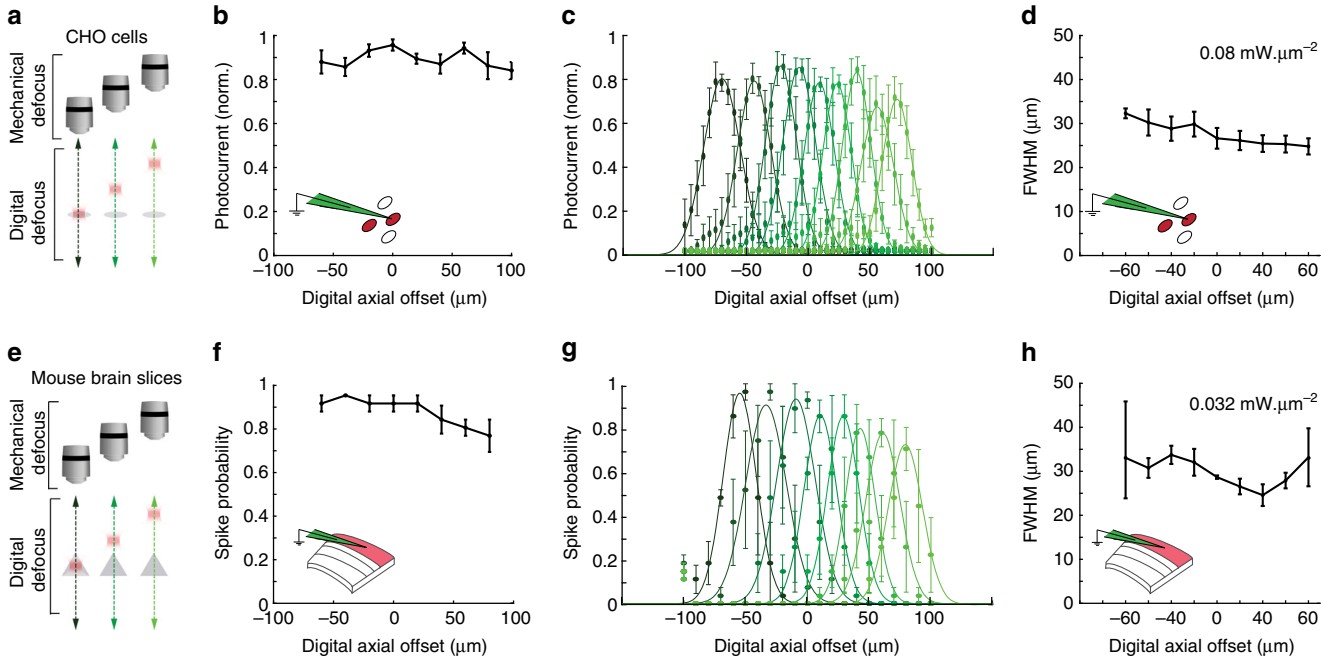

**Fig. 5** 3D-SHOT provides cellular resolution photostimulation in a large volume through digital focusing. **a** To quantify the spatial resolution of 3D-SHOT as a function of hologram target depth, we recorded photocurrents in CHO cells while digitally targeting varying positions along the optical axis ($z$), and measuring resolution by mechanically sweeping the objective over the entire ($z$) range and measuring the response at each point. **b** Normalized photocurrent in CHO cells while targeting the same cell from different axial displacements ($n = 5$ cells; $p = 0.39$, Kruskal–Wallis test with multiple comparisons correction, data are mean and s.e.m.). **c** Axial photocurrent resolution as a function of digital displacement—shaded green colors denote mechanical sweeps across the optical axis for different digital displacements. **d** Quantification of the FWHM for the axial fit of photocurrents in CHO cells as a function of digital defocus from the focal plane ($n = 5$ cells, $p = 0.07$, data are mean and s.e.m.). **e–h** As in **a–d**, but spike probability recorded in mouse brain slices via current clamp instead of photocurrent in CHO cells recorded in voltage clamp (**f**: $p = 0.2$; **h**: $p = 0.17$, $n = 3$ neurons)

combination with our power correction algorithm can deposit equivalent amounts of energy and achieves spatially precise 2P excitation in at least 50 distinct axial planes.

**Control of large volumes with high spatial resolution**. To test the ability of 3D-SHOT to accurately target large numbers of spots in a large volume, we measured 3D-SHOT spatial resolution in numerous contexts. We characterized individual spots within multi-target holograms as a function of increasing the numbers of targets, decreasing the distance between targets, propagation through scattering medium, and ($x,y,z$) position of the target. We generated holograms targeting 20–750 spots spaced 17–135 μm apart, and imaged them through 0–800 μm thick brain slices using high-dynamic range imaging with a sub-stage camera. Using principle component analysis (Supplementary Notes 7, 8), we defined which of these parameters affected the resolution of individual spots within multi-target 3D-SHOT holograms.

The limiting factor for deep brain photostimulation is propagation through brain tissue, where absorption, optical aberrations, and scattering degrade spatial resolution even when operating in the infrared wavelength range. We quantified the effects of optical aberrations on the FWHM by measuring two-photon absorption in 3D through various depths of mouse brain tissue (Supplementary Figs. 7, 8). The effect of scattering on axial resolution was statistically significant after 400 μm (no scatter: $18.1 \pm 5.1$ μm, 400 μm brain tissue: $19 \pm 5.5$ μm, $p = 7.34 \times 10^{-4}$, Kruskal–Wallis with multiple comparisons), but the degradation of the PSF did not exceed 1 μm until 600 μm of mouse brain (600 μm brain tissue: $25.2 \pm 66$ μm). We also observed that performance loss was not only due to a gradual degradation of spatial resolution in deeper layers of the brain, but also to unwanted

deposition of energy outside of the desired volume after passing through about 500 μm of brain tissue. This result defines the maximal operating depth for 3D-SHOT and is similar to other two-photon technologies operating in mouse brain tissue with temporal focusing[47].

To identify the influence of spatial location and target number on spatial resolution independently of the predominant effect of scattering, we conducted additional experiments without scattering tissue. Increasing the number of spots did not significantly degrade the axial resolution until 750 spots were simultaneously illuminated (Fig. 8a, d: 500 spots vs. 600 spots, $p = 0.9$, 600 spots vs. 750 spots, $p < 3 \times 10^{-5}$, Kruskal–Wallis with multiple comparisons). However, we observed a gradually increasing level of off-target two-photon absorption for holograms containing more than 400 targets (Fig. 8d). Since the physical location of each target is known a priori, we computed the contrast ratio by quantifying the amount of two-photon absorption in the targeted neuron-sized volumes, divided by the total two-photon absorption in the entire volume of interest. We estimated that the maximal number of targets in a single hologram to be 380–500, assuming 1–5% background illumination. Contrast further degraded as the number of targets was further increased (Fig. 8e). Notably, this decrease in resolution is not a feature of 3D-SHOT per se, but relates to the overall quality of point cloud holograms, which is limited by the pixel density of the SLM that we employed. Using an SLM with a higher pixel count should allow more targets to be encoded before significant loss of contrast occurs.

Furthermore, we also observed that the FWHM depends on the physical location of targets (Supplementary Fig. 9, Supplementary Note 9). We identified target location along the optical axis to be the dominant factor, $d(FWHM)/dz = 1.8 \times 10^{-2}$, which

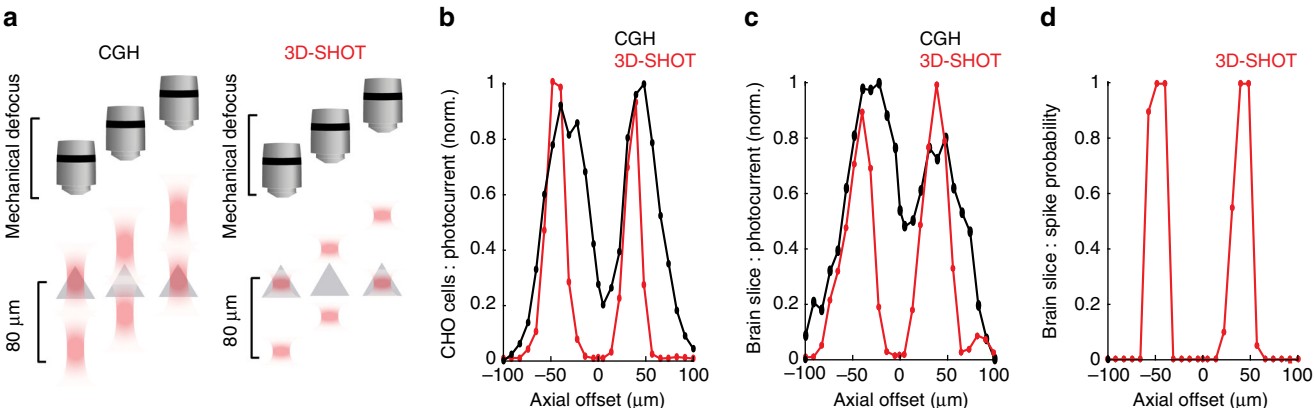

**Fig. 6** Single-neuron resolution with simultaneous photostimulation of two targets along the axial dimension. **a** Using CGH or 3D-SHOT we simultaneously photostimulated two targets separated by 80 µm along the optical z-axis. The photocurrent was recorded as a function of the respective displacement Δz between one patched cell and the volume hologram in an example CHO cell (**b**, n = 11 cells) or in an example cortical neuron (**c**, n = 5 neurons), with conventional holography (black) and with 3D-SHOT (red). **d** As in **c**, but recording spike probability instead of photocurrent for 3D-SHOT

corresponds to known asymmetric geometrical aberrations and magnification properties when attempting to focus light on either side of the natural focal plane of the microscope objective.

We did not observe significant changes in resolution for multi-target holograms as a function of (x,y) position, or as a function of the spacing of spots within a hologram (radial p = 0.17, axial p = 0.29, Kruskal–Wallis with multiple comparisons, Fig. 8f, Supplementary Fig. 9b, d). This independence is bounded by the limiting case in which spots are targeted within one spot-size (specified by the CTFP) of each other, whereupon they constructively interfere and form a continuous area of excitation.

Overall, as with CGH, the performance of 3D-SHOT is limited by either the SLM, the total available laser power, or the microscope system. Altogether, these determine the accessible volume (Supplementary Fig. 4) and the number of neurons that can be simultaneously illuminated with the desired spatial resolution. Here, using an SLM with $600 \times 800$ pixels, we characterized single shot photostimulation of up to 750 targets (limited by laser power) with degradation of resolution within a $0.034 \text{ mm}^3$ volume ($350 \times 350 \times 280 \text{ µm}$, Fig. 8), and improvement of an order of magnitude over previous work using multi-level TF[42].

**Volumetric optogenetics at high spatial resolution**. We next tested the ability of 3D-SHOT to stimulate ensembles of cells distributed in a volume with high spatial resolution. To quantify the PPSF in the context of ensemble stimulation, we generated holograms simultaneously targeting multiple regions of interest distributed throughout the addressable volume and obtained whole-cell recordings from CHO cells while mechanically displacing the multi-target hologram (Fig. 9a). Consistent with direct measurements of 2P absorption, we observed no changes in the lateral (Fig. 9b) or axial (Fig. 9c) PPSF photocurrent FWHM even when simultaneously stimulating 50 regions of interest (p = 0.64 Kruskal–Wallis).

We next tested volumetric optogenetics resolution in neurons by eliciting photocurrents with a single hologram targeting 21 spots, each in a unique z-plane. Instead of mechanically displacing the objective, we computed holograms where only one target was digitally displaced, and the others held in place (Fig. 9d). This approach directly tests the spatial resolution of the target spot while accounting for contamination from non-target spots. These experiments revealed that off-target spots or general loss of contrast do significantly affect photocurrents (Fig. 9e–f). We repeated this experiment for various average power levels,

demonstrating that 3D-SHOT is capable of distributing even very high levels of two-photon light (up to 2 Watts of average power) into spatially confined packets distributed in true 3D throughout a large volume (Fig. 9e–f), and simultaneously (Supplementary Fig. 10).

To validate spatially precise ensemble stimulation in vivo, we used two-photon targeted loose patch in anesthetized mice. As before, we generated holograms that targeted 21 random spots placed throughout a $250 \times 250 \times 200 \text{ µm}^3$ volume, and used the SLM to digitally displace the spot targeting the patched neuron while holding the other spots stationary. In this context, we measured a radial spiking PPSF of $13 \pm 2 \text{ µm}$ and an axial PPSF of $29 \pm 5 \text{ µm}$ (n = 18). These values are very similar to measurements obtained with only one hologram (Fig. 4), and are narrower than the resolution recorded in brain slices due to the additional non-linearity of the action potential threshold (Fig. 9g–i). As predicted from measurements of PSFs through scattering brain slices, there was no relationship between the axial resolution and the depth of the neurons from which we recorded (113–323 µm below the pial surface, Fig. 9j).

Finally, we used a simple model (Supplementary Note 9) to assess expected off-target activation during ensemble stimulation using 3D-SHOT. We randomly placed neurons throughout a $400 \times 400 \times 400 \text{ µm}^3$ volume of brain tissue at physiological density[48] and superimposed 3D Gaussian fits of our PPSF on randomly selected target neurons. This model revealed that even while targeting 500 neurons, on average, non-target neurons are unlikely to be activated (Supplementary Fig. 11a, b). The number of non-target neurons predicted to be active was only a small fraction of the target number, and increasing the number of spots linearly increased the number of activated non-target neurons, indicating that contrast should remain high past 500 spots (Supplementary Fig. 11c). Finally, most of the expected off-target spikes evoked by volumetric 3D-SHOT stimuli resulted from neurons with low spike probabilities, indicating that few off-target neurons should reliably follow repetitive stimulation of a particular neuronal ensemble (Supplementary Fig. 11d).

**Discussion**

In this study, we have demonstrated and validated 3D-SHOT, a method that enables holography and temporal focusing to operate in full 3D with a single SLM. This technology is the first to offer simultaneous targeting of custom neuronal ensembles located anywhere within a large operating volume at cellular resolution,

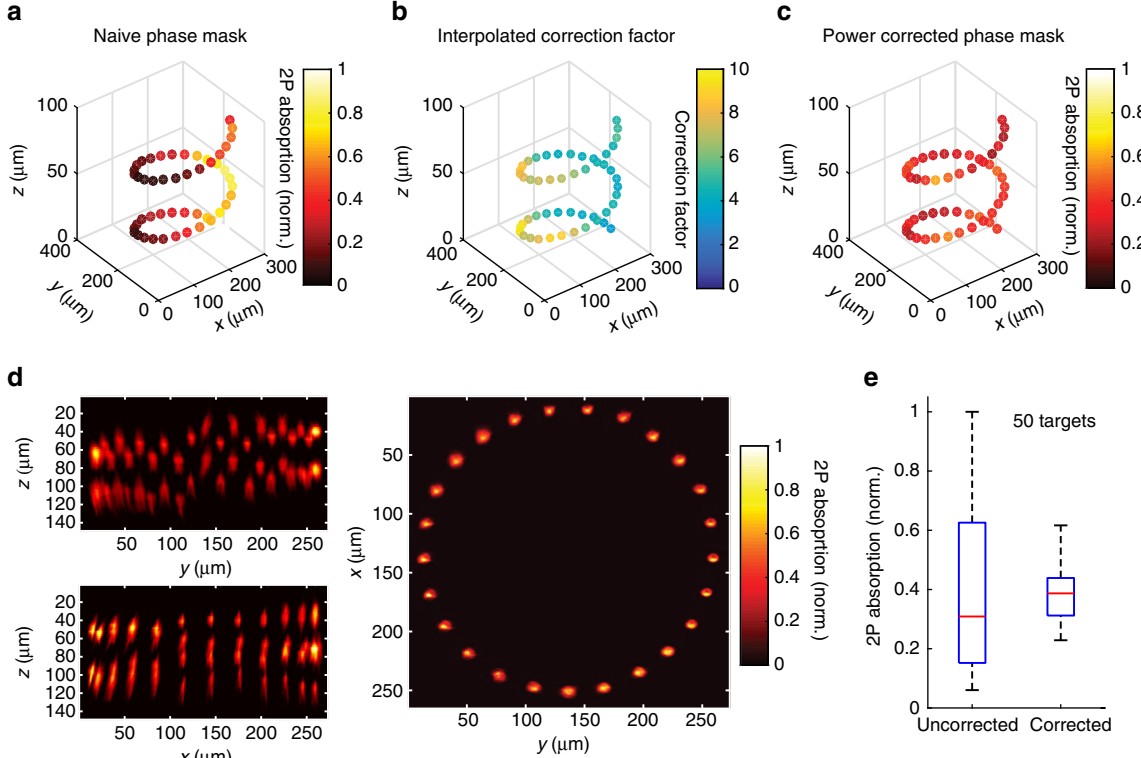

**Fig. 7** Power corrected 3D-SHOT simultaneously illuminates 50 spots. **a** Uncorrected simultaneous 3D-SHOT with a hologram targeting 50 spots in a spiral pattern, occupying 50 individual z-planes. Spots are color coded by normalized 2P absorption in each spot (measured with a sub-stage camera). **b** Correction factor computed for each spot via 3D power interpolant. **c** Measurements of 2P-induced fluorescence with 3D-SHOT, and a power-corrected hologram targeting the same locations as in **a**. **d** Mean intensity images of power-corrected 50 spot spiral hologram showing (z,y) (top left), (z,x) (bottom left), or (x,y) (right) projection views. **e** Box plots showing the variance in normalized 2P absorption from each spot within the corresponding hologram before and after power correction ($p < 1 \times 10^{-12}$, F-test of equality of variances, error bars correspond to range, red line is median, $n = 50$ spots)

and with simultaneous photo-activation of the entire neuron's soma. We have demonstrated the ability of 3D-SHOT to target neurons at high resolution in vivo in mouse brains, and we have shown optogenetic control in a large volume corresponding to the width of multiple cortical columns in mice with single neuron spatial resolution.

Our technology relies on a custom temporally focused pattern (CTFP), which is a fixed-size temporally focused pattern, engineered to enable simultaneous illumination of the entire neuronal cell body, yet confined in a small enough volume to prevent inadvertent photostimulation of non-targeted neurons. We have also engineered the CTFP's phase, which was so far an unused degree of freedom in temporal focusing systems. By patterning the phase, here with a lens ($L_C$), we made holography and temporal focusing compatible by improving the diffraction efficiency at the SLM in a way that allows simultaneous processing of all spectral components of the pulsed light source with a single SLM frame. Spatial resolution in the lateral plane is mostly limited by the numerical aperture of the objective, as in conventional microscopy systems[49]. This criterion is not a significant limiting factor for wide-area targeting of neurons and most future applications in neuroscience.

With conventional holography, defocusing (by simple propagation) is the only available mechanism to attenuate light intensity and confine the nonlinear response near the desired target[20]. Axial resolution being inverse proportional to $NA^2$, depth selectivity requires high-NA objectives, and generally permits single neuron spatial resolution only in small operating volumes[28]. With 3D-SHOT, temporal focusing eliminates this trade-off between spatial resolution and operating volume: cellular resolution along

the z-axis is made possible even with less expensive lower NA objectives, and with smaller magnifications enabling much larger fields of view. The CTFP stretches pulses in time above or below the primary focus, and the nonlinear response decreases with depth regardless of the dimensions of the targeted area in the focal plane. Attenuation of photostimulation away from the targeted neurons is now made possible by the combined effects of physical defocusing and temporal pulse stretching, which are two independent effects, mutually contributing to improving spatial resolution and power stability.

As predicted by simulations and recordings of two-photon absorption in the CTFP, our experiments in CHO cells and neurons show that the physiological spatial resolution, measured in terms of photocurrent, is not only narrower for 3D-SHOT than for conventional holography, but its FWHM is also less sensitive to power variations. This convenient property of the CTFP enables the user to consider conservative estimations of the required power levels to ensure action potentials can be triggered without sacrificing spatial resolution.

Indeed, one ultimate goal for multiphoton photostimulation in brain tissue is to elicit action potentials in many individual neurons with single-cell resolution simultaneously. Recording in voltage clamp, photocurrents linearly reflected the location of the holographic stimuli. When recording action potentials in brain slices or in vivo, the physiological response of a targeted neuron is subject to a non-linearity imposed by action potential threshold. In vivo loose patch recordings and measurement of axial confinement through scattering brain tissue confirm that 3D-SHOT remains efficacious and spatially precise when operating in mouse brains.

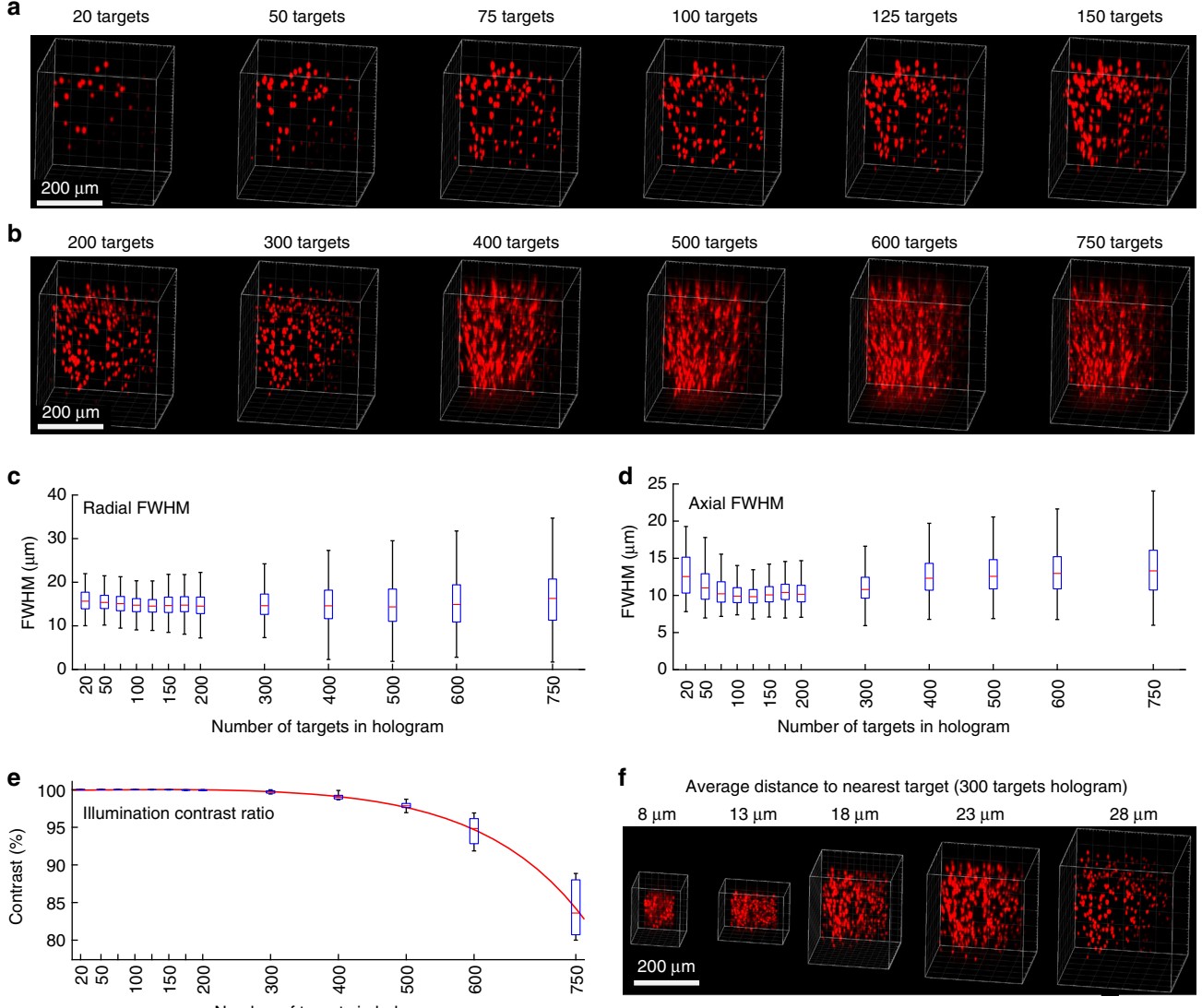

**Fig. 8** Spatial resolution with simultaneous targets throughout a large volume. **a–b** Example 3D renderings from tomographic images of two-photon absorption from single-shot holograms targeting increasing numbers of randomly distributed spots. The FWHMs of the two-photon response was computed for each target, **c** radially, and **d** axially. Results show that spatial resolution and axial confinement are not significantly degraded by increasing the number of simultaneous targets in any given hologram. **e** The contrast ratio of the holograms was quantified as a function of the number of targets, and shows that contrast, rather than resolution determines the total number of targets that can be photostimulated in a single hologram. **f** 3D recordings of various scaling of the same point cloud were made to evaluate density as a possible factor affecting spatial resolution (error bars correspond to range, red line is median)

Using 3D-SHOT, we reliably measured in vivo spiking point spread functions of less than 30 μm axially and about 10 μm radially, even when stimulating up to 21 neurons. As these values closely match the physical size of the soma and proximal dendrites of cortical L2/3 pyramidal neurons, axial PPSFs of about 30 μm have previously been considered consistent with achieving single-neuron spatial resolution[9, 26]. However, while the spatial resolution of patterned optogenetic photostimulation is determined by the optical point spread function, it is also specified by other factors like the subcellular location of opsin on a neuron[50, 51], the density of opsin-expressing neurons, and stimulation frequency and duration convolved with variation in intrinsic neuronal excitability and opsin expression levels. Therefore, in future studies it will be useful to consider single-cell spatial resolution not as a binary switch, but rather, as a continuous variable defined by the probability of generating off-target spikes given by the three-axis physiological point spread function, the number and physical location of other opsin-expressing neurons, and the distributions of opsin-expression and intrinsic excitability. Modeling studies with our physiological point spread function show that even with several hundred targets, while the targeted population is reliably photo-stimulated on each trial, the off-target activated neurons would be similar to very slightly elevated background activity. Notably, this form of analysis of spatial resolution is also relevant for other forms of patterned optogenetic photostimulation.

For photostimulation applications, the current limiting factor for the number of neurons 3D-SHOT can simultaneously control is available laser power and opsin sensitivity, both of which are likely to improve in the future. However, we were able to characterize performance via recordings of two-photon induced fluorescence with a sensitive fluorescent monolayer and camera

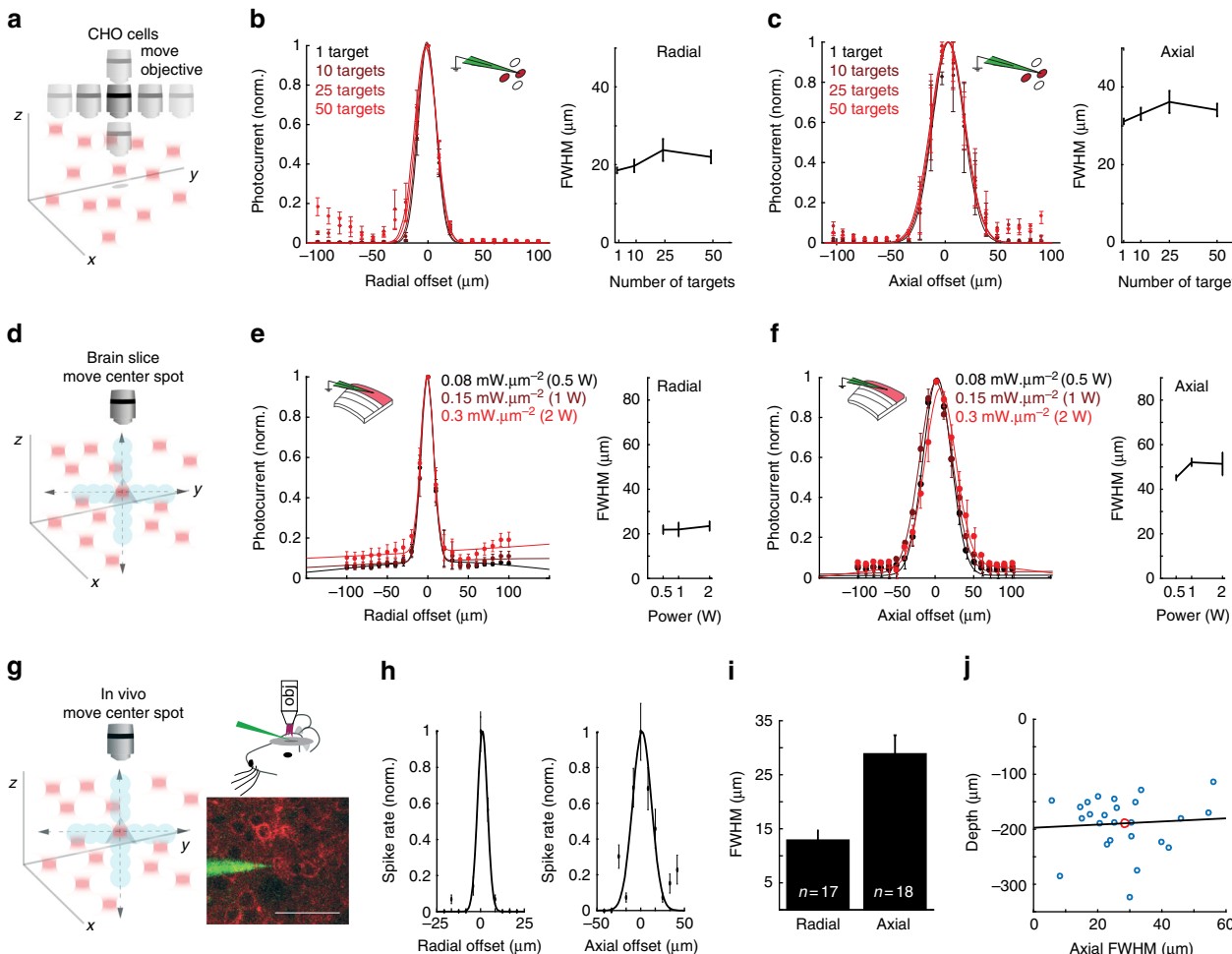

**Fig. 9** Spatially precise volumetric optogenetics with 3D-SHOT. **a** Schematic demonstrating 3D-SHOT stimulation of a CHO cell with a single spot in a hologram targeting multiple spots in the volume. The objective was moved to measure the physiological point spread function via voltage clamp recordings. **b** Normalized photocurrent and quantification of FWHM as a function of radial displacement for holograms targeting 1–50 spots (data represent mean ± s.e.m. of $n = 3$ cells, $p = 0.39$, Kruskal−Wallis test with multiple comparison correction). **c** As in **b** but axially ($n = 3$ cells, $p = 0.64$). **d** Schematic showing whole cell recordings in brain slices while stimulating multiple targets. To measure spatial resolution of 3D-SHOT, a series of new holograms was computed that displaced only the center target while holding all other spots stationary. **e** Photocurrent measurements from brain slices for radial digital offset of a center spot at several stimulation powers (colors, 0.5–2 W), quantification of FWHM as a function of stimulation power through the objective (data are mean ± s.e.m. of $n = 5$ neurons, $p = 0.69$, Kruskal−Wallis test with multiple comparison). **f** As in **e** but axially ($p = 0.11$). **g** As in **d**, but for in vivo loose patch recordings; example image of a 2P guided patch of a L2/3 pyramidal neuron expressing ST-ChrimsonR-mRuby2 (scale bar = 50 μm). **h** Radial and axial physiological point spread functions showing resolution of 3D-SHOT while targeting an ensemble of 21 neurons while only the center spot was moved (data represent mean and s.e.m. from an example neuron). **i** Bar graph showing mean ± s.e.m. of the radial and axial PPSFs from targeted in vivo patch while stimulating 21 cells ($n = 17$ radial and $n = 18$ axial). **j** Scatter plot showing axial FWHM vs. the depth below the cortical surface of individual neurons (axial PPSF is from experiments with 1 or 21 holograms, $n = 24$). Red dot indicates the mean values, and the black line is the line of best fit—within this range, cortical depth can explain none of the variance in axial FWHM ($R^2 = -0.03$)

allowing us to predict that with more powerful laser technology, the current technique will be able to simultaneously activate up to 300–600 neurons per SLM frame with minimal off-target activation, depending on the acceptable levels of contrast between targeted and background volumes. This number is limited by the resolution of the SLM, currently at $600 \times 800$ pixels for 3D hologram synthesis in our system, but higher density SLMs are commercially available with tradeoffs between speed, resolution and diffraction efficiency. The SLM can be chosen to meet the demands of specific neurobiological applications.

In this study, we chose a lens for the CTFP for simplicity, but also because a spherical phase mask provides an analytical expression for the CTFP in space and time. The CTFP dimensions may be adjusted for specific neuronal cell types for brain areas to match the characteristic dimensions of neurons. In specific instances when a circuit is composed of neurons with very different diameters (for example, the cerebellar cortex) a second SLM could be used not only to apply other types of phase patterns to the CTFP to further improve the diffraction efficiency, but also to rapidly change the CTFP dimensions and create custom target shapes in any number of desired 3D locations. However, even with a second SLM, the CTFP will be identical for all stimulation targets at any given moment in time. In the future, the implementation of 3D-SHOT in combination with functional imaging of neural activity will enable real-time manipulation of functionally defined neural ensembles with high specificity in both space and time, paving the way for a new class of experiments aimed at understanding the neural code.

## Methods

**Ethical statement**. All animal experiments were performed in accordance with the guidelines and regulations of the Animal Care and Use Committee of the University of California, Berkeley.

**Two-photon absorption characterization**. CGH and 3D-SHOT characterization experiments were performed by recording two-photon induced fluorescence on a calibration slide with an inverted microscope. We used a Basler ACa2500 camera, and a Leitz 6.3X Objective to map the entire operational range of the SLM on the camera sensor. We placed two infrared filters along the light-path to eliminate the remaining laser light, leaving only the fluorescence signal in the visible range to be recorded by the camera sensor. For power characterization, we used thick auto-fluorescent plastic slides (Chroma) to simultaneously collect fluorescence light within the entire volume of excitation from a single focused image. For more precise 3D characterization of two-photon absorption, we used a custom made thin film of fluorescent paint (Tamiya Color TS-36 fluorescent red) sprayed on a microscope glass slide. Here, by mechanically displacing the image acquisition set-up and the photo-excitation pattern (either from CGH or 3D-SHOT), we recorded two-photon absorption in 3D by digitally assembling slice images captured at various depth levels (with micro-metric mechanical increments). The imaging system was calibrated spatially to confirm the effective magnification of the imaging system, and with single target recordings at known power levels to account for non-uniformity in optical transmission across the field of view of the camera. Up to three recordings at various laser power were made to digitally increase the effective dynamic range of the acquisition process. The FWHM was computed first by identifying isolated targets (threshold-based detection) within multi-target holograms, then by computing projections along any axis of interest and by fitting Gaussian profiles to the resulting data.

**Biological samples preparation**. The cation channelrhodopsins ChrimsonR and Chronos[52]; accession: KF992040.1 generated by gene synthesis (Genewiz, South Plainfield, NJ) and anion opsin GtACR1[53], provided by Dr. John Spudich, University of Texas Health Science Center, Houston) were fused to mRuby2 at the C-terminus at a NotI site (Chronos) or AgeI site (GtACR1) and subcloned into the pCAGGS expression vector between KpnI and XhoI restriction sites by In-Fusion cloning (Clontech, Mountain View, CA). In order to target the opsins to the soma and proximal dendrites of neurons, the sequence encoding the proximal restriction and clustering domain of the Kv2.1 voltage-gated potassium channel consisting of amino acids 536–600[50, 51, 54] was codon optimized, synthesized (Integrated DNA Technologies, Coralville, IA) and inserted at the C-terminus of mRuby2 between BsrGI and XhoI restriction sites by In-Fusion cloning. Chinese hamster ovary (CHO) cells were maintained in Ham's F-12 medium with L-glutamine (Thermo Fisher Scientific, Waltham, MA) containing 10% fetal bovine serum in a humidified incubator at 37 °C and 5% $CO_2$. One day prior to transfection, cells were plated on coverglass (Carolina Biological Supply, Burlington, NC) coated with poly-D-lysine (Sigma-Aldrich, St. Louis, MO) so as to reach a confluence of about 80% and transfected with the expression plasmid encoding either Chronos or GtACR1 using Fugene HD (Promega Corporation, Madison, WI) and recorded 24–48 h post transfection.

**In utero electroporations**. Electroporations were performed on pregnant CD1 (ICR) mice (E15, Charles River ca. SC:022). For each surgery, the mouse was initially anesthetized with 5% isoflurane and maintained with 2.5% isoflurane. The surgery was conducted on a heating pad, and warm sterile PBS was intermittently perfused over the pups throughout the procedure. A micropipette was used to inject ~2 μl of recombinant DNA at a concentration of 2 μg/μl and into the left ventricle of each neonate's brain (typically DNA encoding opsins were doped with plasmids expressing GFP or mRuby3 at a concentration of 1:20 to facilitate screening for expression). Fast-green (Sigma-Aldrich) was used to visualize a successful injection. Following successful injection, platinum-plated 5 mm Tweezertrodes (BTX Harvard Apparatus ca. 45-0489) were positioned along the frontal axis across the head of the neonate with the positive electrode of the tweezers positioned against the left side of the head. An Electro Square Porator (BTX Harvard Apparatus ca. 45-0052) was used to administer a train of $5 \times 40$ mV pulses with a 1 s delay. After the procedure, the mouse was allowed to recover and come to term, and the delivered pups were allowed to develop normally.

**Viral infection**. Neonatal injections were performed as described[55]. P0-4 EMX1-Cre mice were injected with AAV9-CAG-flexed-ST-ChrimsonR-mRuby2 obtained from the UC Berkeley Vision Science Core Gene Delivery Module. Viral aliquots were loaded into a Drummond Nanoject injector. Neonates were briefly cryo-anesthetized and placed in a head mold. With respect to the lambda suture coordinates for S1 injections were 2.0 mm AP, 3.0 mm L, 0.3 mm DV.

**Slice electrophysiology**. We used radial slices from the somatosensory barrel cortex cut along the thalamo-cortical plane or coronal cortical sections. The hemisphere was trimmed on both the anterior and posterior side of barrel cortex with coronal cuts, placed on its anterior side and a cut was made with a scalpel so that much of barrel cortex lay in a plane parallel to cut. The surface of this last cut

was glued to the slicer tray. The preparation was aided by the use of epifluorescent goggles to visualize the expressing area. Two to three 300–500 μm slices were prepared. Cortical slices (400 μm thick) were prepared[56] from the transfected hemispheres of both male and female mice aged P15–P40 using a DSK Microslicer in a reduced sodium solution containing (in mM) NaCl 83, KCl 2.5, MgSO$_4$ 3.3, NaH$_2$PO$_4$ 1, glucose 22, sucrose 72, CaCl$_2$ 0.5, and stored submerged at 34 °C for 30 min, then at room temperature for 1–4 h in the same solution before being transferred to a submerged recording chamber maintained at 30–32 °C by inline heating in a solution containing (in mM) NaCl 119, KCl 2.5, MgSO$_4$ 1.3, NaH$_2$PO$_4$ 1.3, glucose 20, NaHCO$_3$ 26, CaCl$_2$ 2.5.

CHO cells were recorded in the same media, but with the addition of 5–10 μM all-trans-retinal (Sigma-Aldrich, St. Louis, MO). Recordings were performed in either a Cs$^+$ based internal for voltage clamp recordings (CsMeSO$_4$ 135 mM, NaCl 8 mM, HEPES 10 mM, Na$_3$GTP 0.3 mM, MgATP 4 mM, EGTA 0.3 mM, QX-314-Cl 5 mM, TEA-Cl 5 mM) or a potassium gluconate based internal for current clamp recordings (κ-gluconate 135 mM, NaCl 8 mM, HEPES 10 mM, Na$_3$GTP 0.3 mM, MgATP 4 mM, EGTA 0.3 mM). In most experiments, Alexa Fluor 488 or 594 (Thermo-Fisher) was dissolved into the internal solution to enable morphological recovery. Data were analyzed from recordings in which series resistance remained stable and below 30 MΩ. For measuring photocurrent, cells were included if they had holographic currents > 100 pA; for measurements of action potentials cells were included if holographic stimuli could induce spiking within 200 ms. Data were acquired and filtered at 2.2 kHz using a Multiclamp 700B Amplifier (Axon Instruments) and digitized at 20 kHz (National Instruments). All data were acquired using custom written MATLAB (Mathworks) software.

**In vivo patch**. Two-photon guided patch recordings were performed from adult both male and female (35 days or older) wild-type (WT) electroporated adult mice or neonatally injected EMX1-Cre mice (Jackson Labs # 005628). Mice were anesthetized with isoflurane and a custom steel headplate was surgically implanted over the region of interest. A large (2–4 mm diameter) craniotomy was made over the expression site, washed with HEPES ACSF (in mM: NaCl 125, KCl 3, HEPES 10, glucose 10, MgSO$_4$×7 H$_2$O, CaCl$_2$×2 H$_2$O$_2$, set to 300–310 mOsm and pH 7.4) and covered with agarose gel (1%) maintained at 45 °C prior to application. Mice were anesthetized with 1.5 g/kg urethane and 2 μg/kg chlorprothexane before being transferred to recording rig. Body temperature was maintained with a thermal heating pad (FHC) at 37 °C throughout the experiment. Supplementary isofluorane (0–1%) was used to maintain an even level of sedation during recordings, and additional urethane and chlorprothexane was administered every 2–4 h as needed. Cells were identified under a Sutter MOM 2P microscope. Data were acquired using a Multiclamp 700B Amplifier (Axon Instruments) and digitized at 20 kHz (National Instruments). Data was digitally band pass filtered 0.5–2.2 kHz for identification of spikes. All data were acquired using custom written MATLAB (Mathworks) software. Cells were included for analysis if they were spontaneously active and those spikes were sufficiently larger than the noise (>4 Standard deviations of the noise). Furthermore, cells had to pass a 1 P expression check by firing action potentials in response to brief 1 P LED illumination (Lumencore, Sola SE). Cells were holographically stimulated with increasing stimulus durations and laser power until action potentials were reliably generated at 1 Hz stimulation frequency. Spike probability or normalized firing rate was calculated by normalizing the number of action potentials evoked by each stimulus as a function of target position. For in vivo experiments, the minimum number of spikes that occurring during stimulation was subtracted to account for the spontaneous firing rate of the cell.

**Holographic replication of the CTFP by 3D optical convolution**. Let F : $A(x,y) \rightarrow \widetilde{A}(k_x, k_y)$ denote the 2D Fourier transform operator, and $(x,y)$ the lateral coordinates. The relationship between the CTFP field $A_C$ (Supplementary Fig. 1), and the field at the input face of the SLM for each component of the spectrum $\Delta k$ is given by:

$$A_C(x, y, \Delta k) = F\left[A_D^-\right]\left(\frac{x}{\lambda f_4}, \frac{y}{\lambda f_4}, \Delta k\right) \quad (2)$$

We adjusted the phase pattern (focal length of lens $L_C$) at the diffraction grating so that the dimensions of the beam $A_D^-$ matched the dimensions of the SLM's short axis to optimize the number of pixels that can be used for holography (Supplementary Fig. 2).

The SLM applies a phase mask, $\varphi$, and the reflected beam becomes:

$$A_D^+(x, y, \Delta k) = A_D^-(x, y, \Delta k)e^{i\varphi(x,y)} \quad (3)$$

Hence:

$$F\left[A_D^+\right] = F\left[A_D^-\right] \otimes F\left[e^{i\varphi}\right] \quad (4)$$

where $\otimes$ represents the 2D convolution product, and F represents the Fourier transform in the $(x,y)$ plane. Lens $L_5$ applies another optical Fourier transform to build the corresponding hologram, which is further demagnified with the tube lens $L_6$ and the microscope objective $L_7$ to form the final hologram into the brain. The

field $A_E$ is given by:

$$A_E(x, y, \Delta k) = \mathrm{F}[A_D^+]\left(\frac{x}{\lambda f_5}, \frac{y}{\lambda f_5}, \Delta k\right)e^{i\varphi(x,y)} \tag{5}$$

By combining Equations 2, 4, and 5 we find:

$$A_E(x, y, \Delta k) = A_C\left(\frac{f_4 x}{f_5}, \frac{f_4 y}{f_5}, \Delta k\right) \otimes \mathrm{F}[e^{i\varphi}]\left(\frac{x}{\lambda f_5}, \frac{y}{\lambda f_5}\right) \tag{6}$$

We deduce the field in 3D by applying Fresnel propagation to each component of the pulse spectrum:

$$A_E(x, y, z, \Delta k) = A_E(x, y, \Delta k) \otimes h(x, y, z) \tag{7}$$

where $h(x, y, z) = e^{ikz}e^{\frac{ik}{2z}(x^2+y^2)}$ is the Fresnel propagation kernel. The associativity of the convolution product yields the 3D-SHOT principle where the field to be demagnified and imaged into the sample, $A_E(x, y, z, \Delta k)$, is a convolution of the CTFP, $A_C(\frac{f_4 x}{f_5}, \frac{f_4 y}{f_5}, \Delta k)$, and the 3D point cloud hologram object, $\mathrm{F}[e^{i\varphi}](\frac{x}{\lambda f_5}, \frac{y}{\lambda f_5}) \otimes h(x, y, z)$, corresponding to the phase mask, $\varphi$, placed on the SLM.

$$A_E(x, y, z, \Delta k) = A_C\left(\frac{f_4 x}{f_5}, \frac{f_4 y}{f_5}, \Delta k\right) \otimes \left[\mathrm{F}[e^{i\varphi}]\left(\frac{x}{\lambda f_5}, \frac{y}{\lambda f_5}\right) \otimes h(x, y, z)\right]$$

**Identification of factors affecting the FWHM by principal component analysis**. To identify the hologram and brain tissue properties affecting spatial resolution, we used a simple form of principal component analysis (PCA), (Supplementary Notes 7, 8). We consider a diverse data set of multiple holograms for which we gather for each target relevant parameters such as $x, y, z$ location, total number of other targets, brain tissue depth, and distance to the nearest neighbor in a $p$ by $k+1$ matrix, A, (a last column of ones is used to account for a constant offset). From volume recordings of all the holograms, we record either the radial or axial FWHM in a $p$ by 1 data matrix D. We then compute $k+1$ by 1 matrix X that minimizes $\|D - AX\|^2$. With a diverse enough data set, $A \times A$ is invertible and the optimal solution is given by $X = (A \times A)^{-1} A \times D$, The components of X represents the respective contributions of each parameter to the FWHM, with the last one being the constant offset.

**Statistics**. All statistical analysis was performed with Matlab (Mathworks). The tests performed were Mann–Whitney $U$-test for two-sample comparisons, Kruskal–Wallis test with multiple comparisons for multiple sample comparison, and $F$-test of equality of variances. For tests of differences of means, estimates of variance for each group was not typically computed. Each biological replicate is the mean of at least three technical replicates; trial-to-trial variance was extremely low for biological samples. Measurements of fluorescence was performed once per sample. No statistical tests were used to predetermine sample sizes, but we used sample sizes consistent with previous publications[10, 19, 45]. Blinding was not used in this study. Unless otherwise noted, all plots with error bars denote mean ± s.e.m. of the indicated number of biological replicates.

**Data availability**. The data and computer code that support the findings of this study are available from the corresponding author upon reasonable request.

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

## Acknowledgements

We thank Alex Naka for technical assistance. We thank Mei Li and the UC Berkeley Vision Science Core Gene Delivery Module for synthesizing virus. L.W. acknowledges funding from the David and Lucille Packard Foundation and the Moore Foundation. H.A. is a New York Stem Cell Foundation Robertson Investigator and acknowledges support from the Arnold and Mabel Beckman Foundation, the New York Stem Cell Foundation, and from the NIH core grant P30 EY003176. A.M. acknowledges support by NINDS of the NIH under award number F32NS095690-01. I.O. acknowledges the support of the Simon's Foundation Collaboration for the Global Brain award 415569. This work was supported by National Institutes of Health BRAIN R21 Grant EY027597-01 and Defense Advanced Research Projects Agency Contract No. N66001-17-C-4015 to H.A. and L.W.

## Author contributions

N.C.P., L.W., and H.A. developed the principle of 3D-SHOT. N.C.P. assembled experimental device, and performed experimental recordings and simulations for two-photon absorption. A.R.M. and H.A. designed and performed electrophysiology experiments in mouse brain slice and CHO cells. A.R.M. designed and performed in vivo electrophysiology experiments. N.C.P., A.R.M., and I.A.O., wrote code and helped developed software for experimental control. S.S. performed cell culture and transfections. N.C.P., A.R.M., and H.A. wrote the manuscript.

## Additional information

**Competing interests:** The authors declare no competing financial interests.

