## [Peer Review File · Nature Communications]

Reviewers' comments:

Reviewer #1 (Remarks to the Author):

Reviewer's Comments

3D Temporally Focused Holographic Optogenetics at Cellular Resolution

Authors: N. Pegard, A. Mardinly, S. Sridharan, J. Zhang, L. Waller, and H. Adesnik

This manuscript described a novel scanless two-photon excited fluorescence microscopy to capture targeted individual neurons in large volume of tissue by combining temporal focusing and 3D-computer-generated holography. Although the basic idea is very similar to the previous experiment done by Spesyvsev et al., where they combined temporal focusing and holography to capture the fluorescence image of holographically trapped microparticles, the common femtosecond excitation laser pulse, which was temporally focused to generate a 2D-wide field excitation, was holographically shaped in the scheme presented in this manuscript. Proof-of-principle experiment successfully validated the concept and the optical design of this scheme. Therefore, I judged that this manuscript hold a value to be published in Nature Communication. However, before accepting for publication I would like to ask the authors to respond the following mandatory requirements since there are many points puzzling me in the text.

As I mentioned above, the previous paper reported by Spesyvsev et al. (ref. 25) is a significant milestone connecting to 3D-SHOT. Their achievement should be more emphasized in the introduction. Introduction of another spatially shaped temporal focusing scheme reported by Hernandez et al. (ref. 34) was already good enough in the original manuscript.

Since the author used a simple point-focused temporal focusing, the depth resolution \sim (temporal focusing depth) is not high. What is the actual depth resolution of the temporal focusing in this experimental setup? Compared with this depth resolution, what is the interpolation for Δz experimentally obtained, for example for CHO cell?

How did the authors measure spatially resolved photocurrent? It is not clear how the authors obtained the 3D image with the objective lens.

In Fig. 2, the two-photon excited fluorescence image with 3D-SHOT is confined in a smaller volume relative to that generated by 3D-holography-only. The author described that the targeted volume was a 10 μm disk image at $z=0$. How thick is that disk? 10 μm ? or $\Delta z=0$? In that case, does the size of confined fluorescence image in z direction ($\sim 10 \mu\text{m}$) obtained by 3D-SHOT correspond to either the depth resolution of the temporal focusing or the disk thickness?

In Fig. 5, two targets separated by 80 μm along z axis were excited. The focusing depth of the temporal focusing is larger than 80 μm ? How the two spatially separated volume along z axis were simultaneously temporal-focused, if their separation is larger than the temporal focusing depth?

In addition, correct the following errors:

"see Figures 2,2,2,2)"

"substantial photocurrent 50 μm observed above and below....."

Reviewer #2 (Remarks to the Author):

The manuscript entitled „3D Temporally Focused Holographic Optogenetics at Cellular Resolution“ by Nicolas Pegard et al. demonstrates the combination of digital holography with temporal focusing in order to achieve a number of focal spots of cellular dimension for optogenetic stimulation. In this study, the authors introduce a new approach in which an axial shift between the temporal focusing

plane and geometric focus is introduced by applying a spherical phase pattern onto the Gaussian beam illuminating the grating. This allows the authors to multiplex the temporally focused spot within a 3D volume by using an SLM in the Fourier plane.

While interesting I am not convinced that the level novelty represented by this work given previous publications on patterned optogenetics, in particular the work by O. Hernandez et al., Nature Communications 7:11928 (Ref. 34 in the manuscript), is sufficient to warrant publication in Nature Communication. Moreover, there serious issues with how authors discuss state of the art and the body of previous work in the field in the introduction part of their manuscript. Finally, while the authors go a long way is to validate their single-cell resolution claims and characterize parameters such as power dependency, cross-talk in excitation and photo-stimulation the validation of the practical application of the method by which it is motivated, i.e. ontogenetic stimulation of arbitrary spatiotemporal excitation pattern on a large number of neurons and large volumes, is not shown.

Other major points:

- The authors put their focus in the shown experiments on validating the single-cell resolution. This makes their more specialized approach probably more suitable for practical applications albeit it is less versatile compared to the one in Ref. 34. I would like to see the method "in action", i.e. validating the claimed performance of the method while exciting hundreds of neurons within the claimed large volumes.
- Single neuron resolution: As the authors describe this parameter depends on many factors including, expression levels of actuators and power density. The authors support their claims by showing localization of the induced photocurrent as well as spiking probability. While single neuron resolution is difficult to claim for 3D SHOT if FWHM of the induced photo current ($\sim 75\mu\text{m}$) is used (Fig 3e) the authors claim on this point mainly rests on the non-linearity argument that involves when spatial localization of spike probability is used as a measure. However, using spike probability as a metric for spatial resolution is somewhat problematic as the "resolution" in this case will become dependent on how frequently the same neuron is excited. In this case the accumulated sub-threshold currents may eventually lead to spiking. This can be quite import from a practical application point of view, when the same neuron or neuronal assembles are excited repetitively then the "single cell resolution" breaks down.
- Large field of view: The claims on large volume and excitation of many neurons within the volumes within the claimed parameters need to be shown experimentally.
- The paper could benefit from a rigorous discussion and quantitative evaluation of the limits within which the method remains valid. When does it break down? What is the effect of volume size and number of neurons chosen on the obtainable resolution? How would this depend on the amount of introduced shift between the temporal focusing plane and the geometrical focus plane? What are practical limits in a quantitative sense?
- What is the effect of the scattering on this method? It's not clear at what depth the slice experiments were done. If the method should indeed be a useful approach to excited large number of neurons within a large volume. The authors need to show how spatiotemporal excitation and the performance of their method scales as function of tissue depth, ideally in a relevant in vivo setting.
- The manuscript introduction is highly qualitative and in fact until Fig. 3 no numbers are provided on the performance of the system. Below some citations from the text:

"...precisely focuses multiphoton light in space and time and in three dimensions (3D)."

"...to photo-activate arbitrary sets of neurons anywhere within the operating volume".

"...single-neuron spatial resolution of photoactivation in any desired numbers of axial planes"

"...Prior methods such as electrical stimulation or pharmacological manipulation offer either very limited spatial or temporal control"

"single- cell spatial resolution across large volumes remains incompatible with CGH targeting of the entire neuron soma. This issue is especially problematic when low-NA microscope objectives are used to activate cells within larger volumes."

"..to enable single shot in-vivo photo-activation of custom sets of neurons anywhere within a large operating volume and with single neuron spatial resolution".

- The introduction of the manuscript also does not acknowledge previous work in the field, e.g.

"Several methods have been developed for ontogenetic photostimulation, however none are capable of single-neuron spatial and temporal resolution across a large volume." Related methods many of which the authors cite (e.g. Ref 34) methods have been previously published by others but in this context it is also not clear what "large volume" means. This is even more problematic as the authors do not explicitly show how their approach is extending the obtainable volumes and what its ultimate limits would be.

- A large number of references are cited not in a context that is relevant and some relevant citations are missing. Here just a few examples:

- o Ref 19 is not on holography but temporal focusing

- o Ref 23 does not show that "temporal focusing was developed to eliminate this trade off"

- o Ref 24 and 25 seem somewhat arbitrary, there would be many other references including earlier ones on temporal focusing that would make the point

- o The authors claim "... demonstrates how current technology is unable to provide both photoactivation of an entire cell body along with single neuron resolution". This is not correct. Many of the papers that the authors cite (e.g.Ref. 27, 34 and others) have shown single cell resolution optogenetics.

- o Ref 30 is not the first to show combination of mechanical scanning and temporal focusing

- o Ref 33 does not show that multiple depths have been accessed

- The concept of CTFP needs to be more clearly described.

- The authors do not report any details on the used "non-convex nonlinear optimization, this would be necessary

- Many figures are mislabeled which makes reading of the manuscript difficult.

Reviewer #3 (Remarks to the Author):

The manuscript '3D Temporally Focused Holographic Optogenetics at Cellular Resolution' presents a technically interesting - albeit merely incremental - advance in methods for the field of 2P single cell optogenetics.

This reviewer enjoyed reading about the well-designed characterization experiments, which linked the

shape of light confinement to the evoked photocurrents, and to the resulting spiking probability. Nicely done! Such thoughtful characterizations are very informative but oftentimes missing from similar publications.

Computer Generated Holography (CHG), as well as Temporal Focusing (TF), have been applied to 2P photo stimulation of single cells in culture, slice, and in vivo for several years now; temporal focusing was used to improve axial resolution of light patterned using Generalized Phase Contrast (GPC) by the Emiliani lab, and that same lab has recently published the combination of CHG with TF as well, which the authors are aware of since they cited that work (Hernandez et al, Nature Communications, 2016).

Big picture comments:

To differentiate their work from Hernandez et al, the authors state that Hernandez's 'strategy reduces in-plane resolution and is limited to, at most, a few depth levels, still precluding the level of neural control needed for many critical neuroscientific applications.' (1) As far as I can tell, the authors do not provide evidence that their approach is superior with respect to the above stated shortcomings of Hernandez et al. In fact, from Figure S10 of Hernandez et al, it seems like Hernandez et al achieve better axial confinement, even at greater depth and in the presence of more scattering (?) (2) What are the purported 'critical neuroscientific applications' where the differences in performance between this work and Hernandez et al would make a substantial difference?

The first demonstration of opto-optical interrogation of neural circuits in vivo has been reported by the Häusser lab (Packer et al, Nature Methods 12, 140–146 (2015)), followed by Yuste lab's 'Imprinting and recalling cortical ensembles.' (Science, 2016). Even the aforementioned Hernandez paper showed in vivo data (albeit just in zebra fish and not in mammalian tissue). This reviewer feels that the inclusion of in vivo rodent data (which, I believe, the authors are already collecting), showing that the innovations reported actually resulted in tangible benefits for probing the neural code under realistic experimental conditions, would go a long way in justifying the publication of this work in a high impact, general interest journal (rather than in a more specialized, technically oriented venue).

More specific critique:

Abstract:

- '... that can be targeted anywhere within the operating volume': What is the operating volume exactly? There is an implicit claim that said operating volume is large (made explicitly on page 21) compared to prior art, but the data presented for operating volumes of $< 500 \text{ um} \times 500 \text{ um} \times 300 \text{ um}$ appear to be in line with prior art (rather than a significant advancement)
- '... photo activation in any desired number of axial planes': no experimental data has been provided regarding limitations (such as resolution trade-offs) with increasing number of axial planes.
- '... neuroprosthetics to treat brain disease': Typically, neuro prosthetics do not treat brain disease. Rather, they bridge a broken-down biological connection between a healthy brain and external sensors/actuators.

Main text:

- Page 1: Ion channels do not depolarize/hyperpolarize, neurons do.
- Page 4: 'the performance of point-scanning methods remains overall insufficient'. This statement is too general. Why? For what purposes?
- Page 6: ', a necessary condition for interfacing with neural circuits, in vivo': a desirable feature, surely, but 'necessary'? That would depend on the experimental design.
- Page 7: 'cell bodies of neurons tend to have similar shapes and sizes': This is too blanket a generalization. Shapes and sizes vary quite widely depending on genetic identity, location, and

species.

- Page 11: 'non-convex nonlinear optimization': the authors readily admit in the Supplementaries that non-convex optimization offers no convergence guarantees. The authors make the claim that the presented results outperform the more conventional GS algorithm. However, would that result still hold for other arbitrary stimulation patterns of choice?
- Page 11: '(see Figures 2, 2, 2, 2)': abcd missing?
- Page 13: 'The lens Lc, placed into the optical path to make temporal focusing compatible with holography....': Please elaborate for clarity.
- Page 13: 'Each point on the line receives a spectrally narrow pulse that is consequently stretched in time by a factor of ~ 4 ': Please elaborate for clarity
- Page 15: 'The two optical paths are co-aligned so both methods can be tested on individual cells without any digital realignment of the holograms.': In principle, this approach sounds like a great idea for a comparative study. However, I am concerned that the moving mirrors might have somewhat jeopardized alignment. E.g. in Fig (2a), the CGH-spot yz cross-section looks tilted, which may indicate the presence of misalignment and/or other aberrations.
- Pages 17/18: The reported axial confinement for a spot 10 μm in diameter (74 μm) in the xy plane, seems worse than it could be with ideal alignment. For example, this reviewer has measured $\sim 26 \mu\text{m}$ FWHM axially on comparable hardware. Compared to this reviewer's experience with CGH, the reported 3D-SHOT FWHM of 28 μm for a 10 μm (xy) spot does not appear as a significant improvement. Moreover, the claim of 'true single-cell resolution' appears to not be well supported: many neurons in mouse are $< 15 \mu\text{m}$ axially, and even if an adjacent cell does not sufficiently depolarize to produce an action potential reliably, the unintended illumination will significantly bias its spiking probability (which might pose a confound in neural coding experiments).
- Pages 18: '... separated by 80 μm along the optical (z) axis' This choice of separation seems rather arbitrary. It highlights the edge case where the authors' implementation of 3D-SHOT just outperforms their implementation of CGH alone. I am unsure, though, as to how that cherry-picked edge case translates into a tangible benefit under realistic experimental conditions, especially in vivo (?)
- Page 26: '.... refreshing rate ': refresh rate

In summary, I appreciate the authors' detailed, technically oriented manuscript. Once any potentially inflated claims are removed from the abstract, it would be a great fit for a solid optics/engineering journal, in my opinion. However, I would only recommend this paper for publication in a widely read, general interest journal such as Nature Communications, if additional data were provided showing 3D-SHOT optimally aligned, and outperforming the state of the art (Hausser and Yuste papers) in vivo, ideally in an awake behaving animal. At the very least, I would like to see the following questions answered:

- How many cells can be stimulated simultaneously given total available power (1) as a function of depth (2) as a function of xy location in the field of view ?
- What is the (1) spatial and (2) temporal (3) precision (4) reliability of 3D-SHOT photo stimulation in vivo?

Reviewer #1 (Remarks to the Author):

This manuscript described a novel scanless two-photon excited fluorescence microscopy to capture targeted individual neurons in large volume of tissue by combining temporal focusing and 3D-computer-generated holography. Although the basic idea is very similar to the previous experiment done by Spesyvsev et al., where they combined temporal focusing and holography to capture the fluorescence image of holographically trapped microparticles, the common femtosecond excitation laser pulse, which was temporally focused to generate a 2D-wide field excitation, was holographically shaped in the scheme presented in this manuscript. Proof-of-principle experiment successfully validated the concept and the optical design of this scheme. Therefore, I judged that this manuscript hold a value to be published in Nature Communication. However, before accepting for publication I would like to ask the authors to respond the following mandatory requirements since there are many points puzzling me in the text.

We thank the reviewer for their positive comments on our manuscript.

As I mentioned above, the previous paper reported by Spesyvsev et al.(ref. 25) is a significant milestone connecting to 3D-SHOT. Their achievement should be more emphasized in the introduction.

We rephrased our manuscript to emphasize the fundamental difference between our technology and the imaging system developed by Spesyvsev et al. We realize that the potential for confusion comes from stating that we “combine holography and temporal focusing”. **We have rephrased the manuscript to avoid confusion.**

Our technology is distinct from this approach in a number of ways. First, 3D-SHOT combines 3D holography and temporal focusing in a **single optical path**. Our SLM processes femtosecond pulses after they have been spectrally separated by the diffraction grating. Furthermore, our approach employs a custom temporally focused pattern (the CTFP) to impart a phase on the temporally focused beam that hits the SLM. Thus, the custom pattern is replicated at the desired 3D locations specified by the SLM via all-optical convolution. Therefore, while Spesyvsev et al is clearly an important milestone in the use of temporal focusing, it is similar to our study only insomuch as temporal focusing is achieved, and the technologies remain conceptually distinct.

Introduction of another spatially shaped temporal focusing scheme reported by Hernandez et al. (ref. 34) was already good enough in the original manuscript.

We thank the reviewer for this comment. Hernandez et al shows one possible approach for 3D shaping of temporally focused patterns, by combining a diffraction grating and two SLMs on the same optical path. Their strategy offers a possible solution to enable temporally focused photostimulation in multiple axial planes, but not continuous 3D. Reviewers 2 and 3 have specifically asked us to emphasize how our technology is different from Hernandez et al, which

seems to offer a viable avenue towards photostimulation in 3D. **Our technological advance lays in the fact that we do not impose restrictions on the number of depth levels that can be targeted; that is, 3D-SHOT operates continuously in 3D and allows simultaneous targeting of any number of axial planes.** Hernandez et al, however restricts excitation to 4 or 5 axial planes before significant degradation of spatial resolution along the (x,y) directions. This severely limits the addressable neural population when using the approach described in Hernandez et al, making it unsuitable for most neuroscience applications because in most brain structures neurons are located continuously in 3D - not in 4 or 5 discrete depths. Lastly, the technology described in Hernandez et al was not tested with optogenetics, so it is not yet clear if it works in real world scenarios. In contrast, we have further validated 3D-SHOT *in vivo* in mouse (Fig 4).

To demonstrate our continuous 3D targeting capabilities, and thereby more clearly highlight our technical advance over Hernandez et al, we **have introduced additional figures that demonstrates how 50-75 target locations can be illuminated simultaneously with a single phase mask on the SLM at 50-75 different depth levels (Fig 7-8).** Furthermore, we demonstrate the ability to calibrate the energy deposited in each target location, a vital feature for real neuroscience applications (Fig 7). **Finally, we add additional experiments showing that spatial resolution is maintained while simultaneously stimulating living cells with a single phase mask encoding locations at 50 different depths (Fig 9).**

Since the author used a simple point-focused temporal focusing, the depth resolution \sim (temporal focusing depth) is not high. What is the actual depth resolution of the temporal focusing in this experimental setup? Compared with this depth resolution, what is the interpolation for z experimentally obtained, for example for CHO cell?

We apologize to the reviewer for not describing the setup more clearly. In fact, we do not use a simple point-focused temporal focusing. Instead, the laser beam hitting the SLM has phase imparted by a custom temporally focused pattern (CTFP). In this study, this CTFP is a 10 micron disc object and is significantly larger than a diffraction limited spot, although it can be replaced with any desired lens. In first approximation, the temporal focusing depth does not depend on the shape of the illumination pattern¹ on the diffraction grating, but rather on the grating frequency (gamma coefficient in the supplement). In general, the limiting factor to select the appropriate diffraction grating is the back aperture of the microscope objective so as to make sure the full spectrum of the pulse is carried in the temporally focused image. In most TF systems the focusing depth is reduced to a few micrometers². Here, for neural photoactivation, such level of depth confinement is excessive since we wish to activate all the opsin on the neuron soma. Indeed, if the focusing depth is too small, photocurrents can be limited due to incomplete activation of the opsins³.

With regard to the Z interpolation for CHO cells, in our revised manuscript Figure 3 shows this data for various power levels, Figure 5 shows this data as a function of axial displacement, and Figure 9 shows this data as a function of volumetric multi-spot stimulation.

How did the authors measure spatially resolved photocurrent? It is not clear how the authors obtained the 3D image with the objective lens.

We have modified our manuscript to better introduce the methodology for recording spatially resolved photocurrents. Briefly, to measure 3D PSFs, we mechanically moved the objective position relative to the sample and measured the 2P response at each location. These responses thus define the axial PSF. 3D images of the 2P absorption PSFs were recorded using a sub-stage camera acquired in steps (tomographic imaging).

In Fig. 2, the two-photon excited fluorescence image with 3D-SHOT is confined in a smaller volume relative to that generated by 3D-holography-only. The author described that the targeted volume was a 10 μm disk image at $z=0$. How thick is that disk? 10 μm ? or $z=0$? In that case, does the size of confined fluorescence image in z direction ($\sim 10 \mu\text{m}$) obtained by 3D-SHOT correspond to either the depth resolution of the temporal focusing or the disk thickness?

We have rephrased the figure caption to better describe the experimental protocol. The 10 μm disk centered on $z=0$ used for spatial characterization is a single copy of the temporally focused object, and therefore its depth is specified by the depth resolution of the temporal focusing, not the size of the disk.

In Fig. 5, two targets separated by 80 μm along z axis were excited. The focusing depth of the temporal focusing is larger than 80 μm ? How the two spatially separated volume along z axis were simultaneously temporal-focused, if their separation is larger than the temporal focusing depth?

The temporal focusing depth is defined at each conjugate image of the diffraction grating, depending on magnification parameters. However, the introduction of the SLM with point cloud holography allows us to replicate a temporally focused object at different depths. Although it is true that the temporally focused depth may be slightly different depending on the focal depth at which targets are selected, this represents a second-order correction to our model. Thus, each copy of the CFTP, placed 80 μm apart in Z , are independently temporally focused.

In addition, correct the following errors:

“see Figures 2,2,2,2)”

“substantial photocurrent 50 μm observed above and below.....”

We thank the reviewer for pointing out these typos, and we have addressed these issues in the revised manuscript.

Reviewer #2 (Remarks to the Author):

The manuscript entitled „3D Temporally Focused Holographic Optogenetics at Cellular

Resolution” by Nicolas Pegard et al. demonstrates the combination of digital holography with temporal focusing in order to achieve a number of focal spots of cellular dimension for optogenetic stimulation. In this study, the authors introduce a new approach in which an axial shift between the temporal focusing plane and geometric focus is introduced by applying a spherical phase pattern onto the Gaussian beam illuminating the grating. This allows the authors to multiplex the temporally focused spot within a 3D volume by using an SLM in the Fourier plane.

While interesting I am not convinced that the level novelty represented by this work given previous publications on patterned optogenetics, in particular the work by O. Hernandez et al., Nature Communications 7:11928 (Ref. 34 in the manuscript), is sufficient to warrant publication in Nature Communication.

We thank the reviewer for constructive comments. We hope to convince the reviewer that our work contains sufficiently novelty with regards to the study published by Hernandez et al on the strength of three key points.

1) Hernandez et al describes a solution for simultaneous temporal focusing in 4-5 axial planes before degradation of (x,y) resolution⁴. Our solution allows targeting in N arbitrary axial planes - constrained only the microscope objective - and does not suffer breakdown in x,y resolution as more planes are added. Since neurons in most brain structures are distributed continuously in space, targeting in many axial planes is vital for neuroscience applications. **We have added additional data showing simultaneous targeting of 50-75 discrete axial planes, a 15-fold improvement over the state of the art (Fig 7-8).**

2) Our solution is a novel optical system, not a derivative or incremental improvement of the approach described in Hernandez et al. Whereas Hernandez et al uses two SLMs and tiles the Fourier domain into clusters, allowing them to target only 4-5 discrete axial planes before (x,y) resolution degrades to unacceptable levels, our approach uses only one SLM and employs a custom temporally focused pattern (CTFP) to impart phase to the beam that hits the SLM and is then replicated throughout the volume of interest by all-optical convolution.

3) Because our SLM does not need to be tiled we have access to far larger volume of brain tissue. Indeed we show simultaneous illumination of spots over a volume of 350 x 350 x 280 μm , 10-fold better than in Hernandez et al, and many more times better than previous all-optical 2P stimulation studies which operate only in 2D^{5,6}. Again, this study did not test functionality for optogenetic control, whereas we have validated 3D-SHOT in mouse brain, through scattering tissue.

To clarify and highlight the dramatic differences between our work and Hernandez et al, we have included additional figures that demonstrates how our technique can target 50-75 individual locations with a temporally focused pattern at 50-75 separate depth levels, each with the desired level of spatial confinement in a situation where the two-SLM solution proposed in Hernandez et al would not be able to obtain comparable results by tiling the hologram into clusters (Fig 7-9). Therefore, we hope that the reviewer will consider that a 1-SLM method to

achieve true continuous 3D targeting at large scales is a substantial advance over a method limited to 4-5 discrete depths that requires co-alignment of 2 SLMs.

Moreover, there serious issues with how authors discuss state of the art and the body of previous work in the field in the introduction part of their manuscript.

We regret that our original introduction did not accurately present previous works in the field and we thank all the reviewers for pointing out the specific technologies that have been unfairly covered. In this revised manuscript, we have clarified our introduction by better separating the discussion of what technological challenges and desired specifications have motivated the development of 3D-SHOT for specific neuroscience applications, and the presentation of previous work in the field.

Overall, the purpose of this manuscript is to disclose, characterize, and enable easy replication of this novel optical technology. **We have therefore reorganized the experimental results to better focus on spatial performance, accessible volume and included additional experimental results that are relevant to our claim: single shot, simultaneous photoactivation of custom ensembles with single neuron resolution.**

Finally, while the authors go a long way is to validate their single-cell resolution claims and characterize parameters such as power dependency, cross-talk in excitation and photo-stimulation the validation of the practical application of the method by which it is motivated, i.e. optogenetic stimulation of arbitrary spatiotemporal excitation pattern on a large number of neurons and large volumes, is not shown.

We have included new experimental results in the revised manuscript that directly address this issue, and have included new results showing arbitrary spatiotemporal photostimulation patterns (Fig 7-8), and cellular resolution while simultaneously stimulating up to 50 targets in 3D (Fig 9). The supplement also includes a complete presentation of how to achieve proper control of power in individual targets (Fig 7). Additionally, we demonstrate 3D-SHOT *in vivo* in mice using electrophysiology (Fig 4). We hope that this additional data will convince the reviewer that 3D-SHOT is a powerful tool for biological applications.

Other major points:

- The authors put their focus in the shown experiments on validating the single-cell resolution. This makes their more specialized approach probably more suitable for practical applications albeit it is less versatile compared to the one in Ref. 34. I would like to see the method “in action”, i.e. validating the claimed performance of the method while exciting hundreds of neurons within the claimed large volumes.

As stated above, we have included new results showing arbitrary spatiotemporal photostimulation patterns, and cellular resolution in up to 50 targets, in 3D (Fig 7-9). While these experiments demonstrate activation and spatial confinement while stimulating large numbers of targets, using 3D SHOT *in vivo* to activate large ensembles of cells requires many biological advances for expressing soma-targeted opsins powerful enough to generate action potentials.

Using 3D-SHOT *in vivo* to manipulate spatiotemporal neural activity is the subject of another manuscript which focuses on overcoming the biological challenges needed to fully take advantage of the technical advance that 3D-SHOT represents.

- Single neuron resolution: As the authors describe this parameter depends on many factors including, expression levels of actuators and power density. The authors support their claims by showing localization of the induced photocurrent as well as spiking probability. While single neuron resolution is difficult to claim for 3D SHOT if FWHM of the induced photo current (~75um) is used (Fig 3e) the authors claim on this point mainly rests on the non-linearity argument that involves when spatial localization of spike probability is used as a measure. However, using spike probability as a metric for spatial resolution is somewhat problematic as the "resolution" in this case will become dependent on how frequently the same neuron is excited. In this case the accumulated sub-threshold currents may eventually lead to spiking. This can be quite import from a practical application point of view, when the same neuron or neuronal assemblies are excited repetitively then the "single cell resolution" breaks down.

The reviewer raises an important and interesting point, which is that 'single cell resolution' depends on a variety of factors including the frequency of stimulation, the kinetics of the opsin employed, the sensitivity of that opsin to light, the light power needed to activate neurons, and the spatial localization of the opsin on the neuron. Indeed, previous reports in the field of 2P optogenetics have claimed 'single neuron resolution' with an axial spiking PSF of 30 μm ^{7,3}. Therefore, when we use the term 'single neuron spatial resolution,' we refer to a ~30 μm axial spiking PSF, fully understanding the more complex reality, *which depends on many factors distinct from the optical system*.

Since the purpose of this study is to disclose a new optical stimulation paradigm, our goal is to place spatially confined light in a volume consistent with generating a 30 μm spiking PSF. Biologists will no doubt continue to develop more sensitive, fast, and spatially confined opsins to further compliment spatially confined excitation approaches.

Finally, since spatial resolution is intimately related to stimulation power (Fig 3), maintaining single cell resolution will require the ability to precisely control the relative amount of power placed into each spot in a multi-spot hologram. Our updated manuscript now demonstrates our ability to perform such a power normalization, and details how this is achieved in the supplement (Fig 7, Sup Fig 6).

- Large field of view: The claims on large volume and excitation of many neurons within the volumes within the claimed parameters need to be shown experimentally.

We thank the reviewer for highlighting our omission of data demonstrating 3D-SHOT function in a large field of view. To address this we have added two new figures (Fig 7-8) to the manuscript and updated the text to compare the addressable volume using 3D-SHOT to previous published 2P techniques.

- The paper could benefit from a rigorous discussion and quantitative evaluation of the limits within which the method remains valid. When does it break down? What is the effect of volume size and number of neurons chosen on the obtainable resolution? How would this depend on the amount of introduced shift between the temporal focusing plane and the geometrical focus plane? What are practical limits in a quantitative sense?

The amount of spacing between primary and secondary focus is determined by the sphericity of lens L_c and adjusted to ensure best coverage of the SLM to maximize diffraction efficiency. The local characterization of the CTFP then determines the spatial resolution for photostimulation. Independently of this design constraint, our setup remains a standard holographic microscope design with the SLM in Fourier space. Consequently, the accessible volume and number of spots that can be effectively targeted is determined by the optical setup from the SLM to the objective (Supplementary Figure S2), namely the pixel density on the SLM, and the NA of all optical components. In the revised manuscript, the supplement now offers a rigorous quantitative treatment of when the system remains valid or breaks down. The effective of volume and number of neurons is now discussed more quantitatively in the manuscript and is directly addressed by new data in Figures 7-9.

- What is the effect of the scattering on this method? It's not clear at what depth the slice experiments were done. If the method should indeed be a useful approach to excited large number of neurons within a large volume. The authors need to show how spatiotemporal excitation and the performance of their method scales as function of tissue depth, ideally in a relevant *in vivo* setting.

To address this concern, we have conducted additional experiments in which we characterize the spatial resolution of 3D-SHOT excitation when passed through brain slices of varying thickness (Fig 2). This approach allowed us to obtain highly quantitative data on how scattering through varying thicknesses of mouse brain tissue affects spatial resolution. Additionally we validate the method using *in vivo* cell attached patch, demonstrating that the method functions when passed through living mouse brain tissue (Fig 4).

- The manuscript introduction is highly qualitative and in fact until Fig. 3 no numbers are provided on the performance of the system. Below some citations from the text:
 - “...precisely focuses multiphoton light in space and time and in three dimensions (3D).”
 - “...to photo-activate arbitrary sets of neurons anywhere within the operating volume”.
 - “...single-neuron spatial resolution of photoactivation in any desired numbers of axial planes”
 - “...Prior methods such as electrical stimulation or pharmacological manipulation offer either very limited spatial or temporal control”
 - “single- cell spatial resolution across large volumes remains incompatible with CGH targeting of the entire neuron soma. This issue is especially problematic when low-NA microscope

objectives are used to activate cells within larger volumes.”

“..to enable single shot in-vivo photo-activation of custom sets of neurons anywhere within a large operating volume and with single neuron spatial resolution”.

To address this concern, we have updated the text to be more quantitative, citing relevant numbers inline in the text

- The introduction of the manuscript also does not acknowledge previous work in the field, e.g. “Several methods have been developed for ontogenetic photostimulation, however none are capable of single-neuron spatial and temporal resolution across a large volume.” Related methods many of which the authors cite (e.g. Ref 34) methods have been previously published by others but in this context it is also not clear what “large volume” means. This is even more problematic as the authors do not explicitly show how their approach is extending the obtainable volumes and what its ultimate limits would be.

We apologize for the lack of clarity in the manuscript. In the revised manuscript we include new data to demonstrate our system working functionally at the limits of its volume of operation (Fig 8), $350 * 350 * 280 \mu\text{m}$. As noted above, this volume is at least 10x better than that of Hernandez et al. Furthermore, in this calculation, we grant them continuous control of a $300 \mu\text{m}$ axial space, despite the fact that they are limited to 4-5 planes simultaneously. In reality their addressable single-shot volume is substantially lower. As noted above and in the manuscript, the accessible volume is very large compared to previous all-optical studies that stimulate only a single focal plane.

- A large number of references are cited not in a context that is relevant and some relevant citations are missing. Here just a few examples:
 - o Ref 19 is not on holography but temporal focusing
 - o Ref 23 does not show that “temporal focusing was developed to eliminate this trade off”
 - o Ref 24 and 25 seem somewhat arbitrary, there would be many other references including earlier ones on temporal focusing that would make the point

We have addressed the above concerns in the revised manuscript.

- o The authors claim “... demonstrates how current technology is unable to provide both photoactivation of an entire cell body along with single neuron resolution”. This is not correct. Many of the papers that the authors cite (e.g. Ref. 27, 34 and others) have shown single cell resolution optogenetics.

Again we thank the reviewer for their comments, made the appropriate correction with respect to Ref 27 (Papagiakoumou 2010). However, it is very important to note that Ref 34 (Hernandez 2016) does not include any optogenetics experiments (i.e., with microbial opsins). We have revised the manuscript to better reflect the state of the field.

- o Ref 30 is not the first to show combination of mechanical scanning and temporal focusing
- o Ref 33 does not show that multiple depths have been accessed

We thank the reviewers for pointing out these imprecisions and mis-ordered citations, we have revised all our citations and addressed all these issues.

- The concept of CTFP needs to be more clearly described.

We have updated the text to more clearly describe how the CTFP is employed in the 3D-SHOT setup.

- The authors do not report any details on the used “non-convex nonlinear optimization, this would be necessary

We have revised the manuscript to remove reference to non-convex nonlinear optimization. While this approach was fruitful for generating spatially precise holography, we decided that it was ultimately not germane to this study, as GS holography functions well enough for our purposes. Future efforts to improve spatial confinement of 3D-SHOT may employ nonlinear convex optimization, but these efforts will be described elsewhere.

- Many figures are mislabeled which makes reading of the manuscript difficult.

We have addressed this concern by carefully editing the manuscript and ensuring that figure references are properly placed.

Reviewer #3 (Remarks to the Author):

The manuscript ‘3D Temporally Focused Holographic Optogenetics at Cellular Resolution’ presents a technically interesting - albeit merely incremental - advance in methods for the field of 2P single cell optogenetics.

This reviewer enjoyed reading about the well-designed characterization experiments, which linked the shape of light confinement to the evoked photocurrents, and to the resulting spiking probability. Nicely done! Such thoughtful characterizations are very informative but oftentimes missing from similar publications.

We greatly appreciate the reviewer’s kind comments.

Computer Generated Holography (CHG), as well as Temporal Focusing (TF), have been applied to 2P photo stimulation of single cells in culture, slice, and in vivo for several years now; temporal focusing was used to improve axial resolution of light patterned using Generalized Phase Contrast (GPC) by the Emiliani lab, and that same lab has recently published the combination of CHG with TF as well, which the authors are aware of since they cited that work

(Hernandez et al, Nature Communications, 2016).

Big picture comments:

To differentiate their work from Hernandez et al, the authors state that Hernandez' 'strategy reduces in-plane resolution and is limited to, at most, a few depth levels, still precluding the level of neural control needed for many critical neuroscientific applications.' (1) As far as I can tell, the authors do not provide evidence that their approach is superior with respect to the above stated shortcomings of Hernandez et al. In fact, from Figure S10 of Hernandez et al, it seems like Hernandez et al achieve better axial confinement, even at greater depth and in the presence of more scattering (?) (2) What are the purported 'critical neuroscientific applications' where the differences in performance between this work and Hernandez et al would make a substantial difference?

As we note in above responses, the significant improvement in 3D-SHOT over the work of Emiliani and Hernandez is the ability to target continuously in 3D, with no limit on the number of simultaneously addressable axial planes. In neuroscience, most brain structures (including the cerebral cortex, where the vast majority of 2P optogenetics studies are performed), neurons are positioned continuously - not discretely - in axial space. Therefore the ability to only simultaneously address 4 axial planes *critically limits* the combinations of neurons that can be simultaneously activated, placing severe constraints on the promise of scanless 2P optogenetics to parse the neural code.

To further illustrate our advance we have included new figures demonstrating our ability to hit 75 targets simultaneously in 50-75 axial planes without loss of (x,y) resolution (Fig 7-8). Furthermore, our approach is a novel optical design only SLM, not 2, as in the Hernandez study, and therefore is not an incremental advance, but a novel solution that we demonstrate to perform at least 15 times (75 vs 5 axial planes) better than the current state of the art.

Furthermore, it is vital to note that the Hernandez study employed a 60x objective, whereas we use a 20x objective. This allows our accessible volume to be significantly larger (9x larger), another important feature for neuroscience applications. Finally, the confinement of light reported the Hernandez study is optical data and from larval zebrafish, which has almost no scattering. In contrast, we rigorously characterize the photocurrent response to 3D-SHOT excitation across multiple axial depths (Fig 5), with two stacked targets (Fig 6), *in vivo* (Fig 4), and in the context of large scale volumetric stimulation (Fig 9). Since the Hernandez study did not apply their method to actually generating photocurrents even in mouse brain slices, it is very difficult to ascertain whether they are indeed achieving better axial confinement (even using a 60x compared to a 20x objective) of photoexcitation, or whether their approach is useful for optogenetics studies in mouse.

The first demonstration of opto-optical interrogation of neural circuits *in vivo* has been reported by the Häusser lab (Packer et al, Nature Methods 12, 140–146 (2015)), followed by Yuste lab's 'Imprinting and recalling cortical ensembles.' (Science, 2016). Even the aforementioned

Hernandez paper showed *in vivo* data (albeit just in zebra fish and not in mammalian tissue). This reviewer feels that the inclusion of *in vivo* rodent data (which, I believe, the authors are already collecting), showing that the innovations reported actually resulted in tangible benefits for probing the neural code under realistic experimental conditions, would go a long way in justifying the publication of this work in a high impact, general interest journal (rather than in a more specialized, technically oriented venue).

We thank the reviewer for this comment and rephrased our manuscript to accurately introduce the respective contributions of these groups in the development of all optical interrogation techniques. We have included new data to further strengthen the key claims of our new technology: single-cell body confinement of photoexcitation *in vivo* (Fig 4), and custom targeting of whole cells in 3D within the spatial range of conventional holographic methods (Fig 7-9).

We have rephrased our manuscript to emphasize the technological aspects of 3D shot we wish to share with the community of researchers in optics and neurosciences, and we have conducted additional *in vitro* experiments and characterizations that directly tackle the reviewers' concerns regarding our ability to simultaneously activate multiple areas with the desired level of spatial confinement and within the claimed accessible volume (Fig 7-8).

We also show additional characterization experiments to demonstrate robustness to optical scattering and aberrations (Fig 2). However, since the focus of this paper is to describe the development and use of a novel optical stimulation paradigm, we strongly feel that demonstrating all-optical interrogation of the neural code, as in the Yuste and Hausser papers, are beyond the scope. Indeed, we note that report of the optical stimulation paradigm used in the Yuste and Hausser papers preceded those studies and was reported in independent publications^{8,9}.

More specific critique:

Abstract:

- '... that can be targeted anywhere within the operating volume': What is the operating volume exactly? There is an implicit claim that said operating volume is large (made explicitly on page 21) compared to prior art, but the data presented for operating volumes of < 500 μm x 500 μm x 300 μm appear to be in line with prior art (rather than a significant advancement)

To address this concern, we provide new data demonstrating our ability to target 75 spots in 75 planes across a volume of 350*350*280 μm (Fig 8). We also provide a brief comparison in the text comparing this volume to the state of the art (as defined by the three studies mentioned above by the reviewer). Furthermore, a complete discussion of the relevant parameters that define the accessible volume is now available in the supplement. We confirm that the available accessible volume is in line with prior art, and compares, by design, to the accessible volume for any holographic system. Here, the significant improvement is that temporal focusing allows us to offer single neuron resolution without placing restriction on the accessible volume.

With computer generated holography, whole cell photostimulation and single neuron spatial resolution are only compatible in a trade-off that is set by the objective NA. Since defocusing is the only mechanism that allows for rapid decrease of the intensity away from a given target, single neuron resolution requires high NA, and performance rapidly degrades away from the optical axis. For photostimulation, the relevant performance indicator is the accessible volume for which targeting can be achieved with single neuron resolution. We have designed our microscope to reach a specific volume and demonstrated that in such conditions, CGH fails at providing single neuron resolution capabilities, and justified the need for an advanced strategy.

- '... photo activation in any desired number of axial planes': no experimental data has been provided regarding limitations (such as resolution trade-offs) with increasing number of axial planes.

To address this concern, we have added a new data figure, described above, that demonstrates single-shot simultaneous targeting of 50 spots at 50 distinct axial depths (Fig 7) and 75 spots at 75 depths (Fig 8). Additionally, the manuscript has been updated to address the limit cases and resolution tradeoffs for adding additional depths.

- '... neuroprosthetics to treat brain disease': Typically, neuro prosthetics do not treat brain disease. Rather, they bridge a broken-down biological connection between a healthy brain and external sensors/actuators.

Main text:

- Page 1: Ion channels do not depolarize/hyperpolarize, neurons do.

- Page 4: 'the performance of point-scanning methods remains overall insufficient'. This statement is too general. Why? For what purposes?

- Page 6: ', a necessary condition for interfacing with neural circuits, in vivo': a desirable feature, surely, but 'necessary'? That would depend on the experimental design.

- Page 7: 'cell bodies of neurons tend to have similar shapes and sizes': This is too blanket a generalization. Shapes and sizes vary quite widely depending on genetic identity, location, and species.

- Page 26: '.... refreshing rate ': refresh rate

We have revised the manuscript to address these concerns.

- Page 11: 'non-convex nonlinear optimization': the authors readily admit in the Supplementaries that non-convex optimization offers no convergence guarantees. The authors make the claim that the presented results outperform the more conventional GS algorithm. However, would that result still hold for other arbitrary stimulation patterns of choice?

We have eliminated non-convex optimization from this manuscript, as GS functions adequately, and the optimization is not strictly relevant to the design and implementation of 3D-SHOT.

- Page 11: '(see Figures 2, 2, 2, 2)': abcd missing?
- Page 13: 'The lens Lc, placed into the optical path to make temporal focusing compatible with holography...': Please elaborate for clarity.
- Page 13: 'Each point on the line receives a spectrally narrow pulse that is consequently stretched in time by a factor of ~4': Please elaborate for clarity

We thank the reviewer for their careful reading of the manuscript and have edited the manuscript to address these concerns.

- Page 15: 'The two optical paths are co-aligned so both methods can be tested on individual cells without any digital realignment of the holograms.': In principle, this approach sounds like a great idea for a comparative study. However, I am concerned that the moving mirrors might have somewhat jeopardized alignment. E.g. in Fig (2a), the CGH-spot yz cross-section looks tilted, which may indicate the presence of misalignment and/or other aberrations.

The tilt observed in Figure 2a is a feature related to the microscope objective. Most commercial objectives are geometrically compensated to provide focused images in the focal plane (located at the specified working distance from the bottom of the objective), but this correction is generally invalid anywhere else above or below the focal plane. In 3D applications, we expand the operational range to build holographic and 3D-SHOT patterns that extend far above and below the operating depth for which it was initially designed in a regime where geometrical aberrations become significant and induce the observed tilt. Regardless of the detectable tilt, alignment is acceptable to ensure proper targeting, as long as the beam in location "C" (a focused Gaussian spot with CGH or a copy of the CTFP with 3D-SHOT depending on the sliding mirror setting), are superimposed, with their respective centers in "C" co-aligned in all X,Y,Z directions. Also, since CGH and 3D-SHOT undergo the same geometric aberrations, holograms can be computed in identical SLM-based system of coordinates and will target the same physical locations. The objective-induced aberrations can be digitally compensated if necessary by performing a spatial calibration.

- Pages 17/18: The reported axial confinement for a spot 10 μm in diameter (74 μm) in the xy plane, seems worse than it could be with ideal alignment. For example, this reviewer has measured ~ 26 μm FWHM axially on comparable hardware. Compared to this reviewer's experience with CGH, the reported 3D-SHOT FWHM of 28 μm for a 10 μm (xy) spot does not appear as a significant improvement. Moreover, the claim of 'true single-cell resolution' appears to not be well supported: many neurons in mouse are <15 μm axially, and even if an adjacent cell does not sufficiently depolarize to produce an action potential reliably, the unintended illumination will significantly bias it's spiking probability (which might pose a confound in neural coding experiments).

The reviewer suggests that axial confinement for a 10 μm spot using CGH (74 μm) is excessive, and references a 26 μm FWHM that they measured. Without additional information about the specific experiment that the reviewer is referring to, it is hard to know exactly how to explain this

discrepancy. While we do not doubt the veracity of the reviewer's statement, we also do not doubt the fact that our system is extremely well aligned.

Several factors may account for the difference between the reviewer's value of 26 μm , and our measured value of 74 μm .

1) Response Medium: the 74 μm FWHM is obtained by measuring the 'physiological point spread function (PPSF)' in neurons. This PPSF is the interaction of the light with the opsin present on the neuronal processes, and therefore the PPSF will always be substantially larger than the true light confinement, since it represents the convolution of the light with the size of the excitable membrane

2) Power Levels: as we show in Figure 3, the FWHM measured using CGH is more sensitive to the power levels used for stimulation than 3D-SHOT. The 74 μm value was recorded at the highest power levels; since neurons can actually be induced to spike at low power levels (for instance by using a longer stimulus duration), a functional FWHM could be lower in practice. Therefore, the functional spatial resolution depends heavily on the stimulation duration and kinetics of the opsin - this is the subject of upcoming manuscript entirely focused on the neurobiological applications of 3D shot.

3) Objective: Here, we use a 1 NA 20x objective, since this is the objective needed to perform biological experiments that cover a large accessible volume. The FWHM using CGH can indeed be lowered by using higher magnification, higher NA objectives, but they do not represent standard use-cases for biological experiments, and would indeed severely restrict the addressable volume. Similar papers, for instance the Herndandez study noted above, employ a 60x objective to measure PSFs. In this study, most experiments were performed using a 20x objective needed for biological experiments. While this can result in apparently worse measurements, we believe that is important to measure these values in the same context as their expected use-cases.

Finally, the reviewer raises an excellent point regarding biasing nearby neurons to spike. In fact, whether or not one has true single cell spatial resolution is an extremely complex condition to assess: rather than a yes or no, this is probably best defined by a continuous variable defined as the probability of generating an off target spike given by a model accounting for variables such as: stimulation power, rate, duration, opsin kinetics, neuronal rheobase, location of nearby opsin-expressing cells, variability in opsin expression, etc. Thus, single neuron resolution can only be assessed in a combined optical and biological context. Since the purpose of this manuscript is to describe a novel optical paradigm, we only seek to demonstrate that we generate confined light consistent with the possibility of gaining single neuron resolution. In other 2P optogenetics studies this is a term of art describing an axial spiking FWHM of $\sim 30 \mu\text{m}$ ^{7,3}.

- Pages 18: '... separated by 80 μm along the optical (z) axis' This choice of separation seems rather arbitrary. It highlights the edge case where the authors' implementation of 3D-SHOT just outperforms their implementation of CGH alone. I am unsure, though, as to how that

cherry-picked edge case translates into a tangible benefit under realistic experimental conditions, especially *in vivo* (?)

To address this concern we have updated the manuscript by clearly motivating the choice of 80 μm . Indeed, since average cortical neurons have size 25-30 μm axially, 80 μm was chosen to represent the closest packing where one might wish to stimulate two neurons located directly above and below a given cell with NO (x,y) displacement, based on typical neuron density in cortical layer 2/3.

To further address the reviewers concerns, we added additional experiments performing *in vivo* cell attached patch measurements of the spiking physiological point spread function, and demonstrate the 3D-SHOT significantly outperforms CGH in mouse brain tissue (Fig 4).

In summary, I appreciate the authors' detailed, technically oriented manuscript. Once any potentially inflated claims are removed from the abstract, it would be a great fit for a solid optics/engineering journal, in my opinion. However, I would only recommend this paper for publication in a widely read, general interest journal such as Nature Communications, if additional data were provided showing 3D-SHOT optimally aligned, and outperforming the state of the art (Hausser and Yuste papers) *in vivo*, ideally in an awake behaving animal. At the very least, I would like to see the following questions answered:

Based on reviews, we have provided significant additional data to document our claims in terms of performance. We have also toned down the abstract and manuscript to accurately represent our data.

- How many cells can be stimulated simultaneously given total available power (1) as a function of depth (2) as a function of xy location in the field of view ?

To address these questions we have added several additional data figures showing 2P stimulation of CHO cells and neurons in the context of multiple targets distributed throughout the volume of interest. We demonstrate in figures 7-8 that 50-75 spots can be targeted throughout a volume of 350 * 350 * 280 μm field of view. In figure 9 we show that confinement of excitation is unaffected even when 50 targets are simultaneously hit. Additionally, we demonstrate that the ability to activate neurons with high spatial resolution is unaffected by depth in Figure 5.

- What is the (1) spatial and (2) temporal (3) precision (4) reliability of 3D-SHOT photo stimulation *in vivo*?

To address these concerns:

1) We have added an *in vivo* characterization of the spatial resolution of 3D-SHOT as compared to CGH (Fig 4: 70 μm vs 30 μm axial FWHM).

2) The temporal components of 3D shot are limited only by the time it takes to open a pockels cell (microseconds) or the refresh rate of an SLM; ours updates at 30 Hz, but commercially available SLMs update at 500 Hz. Therefore the temporal resolution of 3D-SHOT stimulation is limited only by the commercially available hardware, not the optics. We have added a supplemental figure to demonstrate this point (Sup Fig 7).

3-4) precision and temporal reliability: we understand the reviewer to refer to the precise timing of spikes elicited by 3D-SHOT and the probability of generating a spike using 3D-SHOT *in vivo*. Since these parameters are defined by the intrinsic properties of the neuron and the strength and kinetic properties of the opsin, they have nothing to do with 3D-SHOT *per se*, and are therefore beyond the scope of this study. However, using conventional ultra-fast opsins we have achieved sub-millisecond temporal precision using 3D-SHOT stimulation, and this work has been submitted as part of a separate manuscript devoted to opsin-engineering.

Works Cited

1. Dana, H. & Shoham, S. Numerical evaluation of temporal focusing characteristics in transparent and scattering media. *Opt. Express* **19**, 4937–48 (2011).
2. Spesyvtsev, R., Rendall, H. A. & Dholakia, K. Wide-field three-dimensional optical imaging using temporal focusing for holographically trapped microparticles. *Opt. Lett.* **40**, 4847–4850 (2015).
3. Papagiakoumou, E. *et al.* Scanless two-photon excitation of channelrhodopsin-2. **7**, (2010).
4. Hernandez, O. *et al.* Three-dimensional spatiotemporal focusing of holographic patterns. *Nat. Commun.* **7**, 1–10 (2016).
5. Packer, A. M., Russell, L. E., Dagleish, H. W. P. & Häusser, M. Simultaneous all-optical manipulation and recording of neural circuit activity with cellular resolution *in vivo*. **12**, (2014).
6. Carrillo-reid, L., Yang, W., Bando, Y., Peterka, D. S. & Yuste, R. Imprinting and recalling cortical ensembles. *Science (80-.)*. **353**, 691–694 (2016).
7. Nikolenko, V., Poskanzer, K. E. & Yuste, R. Two-photon photostimulation and imaging of neural circuits. *Nat. Methods* **4**, 943–950 (2007).
8. Packer, A. M. *et al.* Two-photon optogenetics of dendritic spines and neural circuits. *Nat. Methods* **9**, 1202–5 (2012).
9. Rickgauer, J. P., Deisseroth, K. & Tank, D. W. Simultaneous cellular-resolution optical perturbation and imaging of place cell firing fields. *Nat. Publ. Gr.* **17**, 1816–1824 (2014).

Reviewers' comments:

Reviewer #1 (Remarks to the Author):

[only submitted comments to the editor]

Reviewer #2 (Remarks to the Author):

Report:

In their revised manuscript, the authors have addressed many of the points that I have raised in my first report. However, several concerns as outlined below remain which prevent me to recommend the manuscript for publication. The authors respond:

We have included new experimental results in the revised manuscript that directly address this issue, and have included new results showing arbitrary spatiotemporal photostimulation patterns (Fig 7-8), and cellular resolution while simultaneously stimulating up to 50 targets in 3D (Fig 9). The supplement also includes a complete presentation of how to achieve proper control of power in individual targets (Fig 7). Additionally, we demonstrate 3D-SHOT in vivo in mice using electrophysiology (Fig 4). We hope that this additional data will convince the reviewer that 3D-SHOT is a powerful tool for biological applications.

However, the actual and the expected limits of the performance of the method is still not demonstrated in a convincing way but the authors seem rather to choose the optimal conditions for showing their results. I consider this misleading. I believe the reader would appreciate an unbiased representation of the limits at which the method (like any other method) will start to fail. This by no means would devalue the merits of their method but would rather provide a more rigorous and objective characterization. Here are some specific points:

Fig 8: The authors show the axial and lateral resolution in within a volume as a function of ROIs per hologram. What is more interesting is the lateral and spatial resolution as a function of depth (z). How is that for a population of neurons at 300um or 400um depth?

The authors show activation of 75 targets only. However, there must be more neurons within the shown volume than 75. Why can't they activate all neurons within that volume? And how would the spatial resolution look like if they activated 80 neurons that are not distributed far from each other in 3D in such a volume but rather within close proximity of each other, i.e. at the average distance of individual neurons in cortex?

Across ROIs it seems that the axial FWHM is consistently lower (by about 10um) than the radial one. Why? And how can the authors justify that this is sufficient for activation of single neurons? I would expect that this would be even larger if neurons at 400um depth would be evaluated.

Fig9: Again, it's not clear at what depth the results were obtained. If authors claim single neuron resolution of activation how is it justified that as shown in f) a FWHM of $\sim 50\mu\text{m}$ – not considering the depth at which the data was taken – still constitutes single neuron resolution?

Also, here again only 50 targets were chosen, thus the same arguments and questions as above apply.

On a more general level, and this goes back to my comments in my previous report on the

demonstration of the method in vivo, the authors need to convince the biological users that this is a method that can be used for actual biological studies and not a just a proof of principle. Thus, the authors need to show their method under actual conditions in which the method will be used, i.e. in vivo, at some reasonable depth, i.e. layer 2/3 neurons at least, for a sufficient large number of neurons and then show what are the conditions at which the method breaks down, i.e. at what depth, at what number spots, etc.

If the intention of the authors is just to show proof of principles and the optical innovation then, I think a more specialized applied optics journal might be the better medium. In this context the authors seem to contradict themselves in their response: They say : "...have motivated the development of 3D-SHOT for specific neuroscience applications..." while at the same time the claim "Using 3D-SHOT in vivo to manipulate spatiotemporal neural activity is the subject of another manuscript which focuses on overcoming the biological challenges..."

As stated in my previous report, to be convincing the authors need to show their method in action, i.e. to demonstrate the performance of their method and their claims in vivo in an unbiased fashion. Instead currently the authors pick individual aspects of their claim and show them under somewhat idealized conditions or while ignoring the effect on other parameters. Again, the method should be seen through the eyes of an end-user. There is a serious investment required to setup such a system in a systems neuroscience lab and ultimately the more convincing the authors can be to the end-users the higher will be the impact of their technique.

Reviewer #3 (Remarks to the Author):

The authors have addressed many of my concerns through changes to the manuscript, and additional clarification in their rebuttal letter, adequately. I was particularly excited about the characterization of the point-spread function of L2/3 neurons via 2P guided patch recordings in vivo.

Provided that some minor issues are corrected, as outlined below, I would feel more comfortable now recommending this manuscript for publication in Nature Communications.

Introduction:

1. The authors' claim in several locations in the manuscript (e.g. in the abstract) that their technique enables photoactivation 'in any number of focal planes'. In my understanding, the actual claim should be that the focal plane can be chosen from a continuum such that each stimulation location can lie in a unique focal plane if need be. It should be a statement about the unrestricted choice of stimulation planes (rather than about their unlimited total number).

2. The following claim in the intro is still overstated, please narrow it to reflect the actual advance this work is showing:
'Several methods have been developed for optogenetic photostimulation, however none are capable of simultaneous single neuron spatial resolution [...] while offering temporal precision across a large volume.'

3. Modify/remove 'uses absorption in the visible spectrum to directly activate the opsin' from the description of 1P optogenetics. This is misleading as it implies that somehow 1P processes are direct and 2P processes are indirect, which is not a canonical use of terminology.

4. 'Since two-photon absorption is a non-linear process proportional to the square of the light intensity' - not the process is proportional but its efficiency or probability etc.
5. I would write 'achieve/sustain photocurrent' rather than 'accumulate'
6. No need to write 'coherent laser beam' : laser beams are coherent by definition
7. In the explanation of temporal focusing 'depth of this layer' should be replaced by 'thickness of this layer'.

Selective photostimulation of vertically stacked neurons:

- The statement that the typical z-extent of a 'cortical pyramidal neuron is 25-30 μm ' is incorrect, at least as stated. Please specify species, layer, cortical area, and provide a reference.

Simultaneously addressing arbitrary 3D locations...

- The authors use the term 'tomographic' to denote z-slices. Tomography typically refers to z-slices reconstructed from line-integrals via inverse Radon transform. This is not the case here, so please just say z-slice in lieu of 'tomographic image'.

Discussion:

- entire neuron's soma instead of 'entire neurons' soma'
- replace 'to spike neurons' by 'to activate neurons', 'elicit action potentials' or something similar.
- In (not 'with') CHO-cells [...] underlying photocurrent [...] linearly reflects the holographic patterns: Please rephrase this sentence to increase clarity. I understand that you wish to contrast the presence of just opsins in CHO vs the additional presence of voltage-sensitive channels in real neurons - just make this more clear & explicit.
- Even with a second SLM, the CTFP created will be identical for all stimulation locations at a given point in time. This should be stated more clearly at the end of the discussion.

Please find below a point-by-point response specifically addressing the remaining concerns expressed by reviewers.

Reviewer #1

We thank the reviewer for evaluating our manuscript.

Reviewer #2 (Remarks to the Author):

In their revised manuscript, the authors have addressed many of the points that I have raised in my first report. However, several concerns as outlined below remain which prevent me to recommend the manuscript for publication. The authors respond:

We have included new experimental results in the revised manuscript that directly address this issue, and have included new results showing arbitrary spatiotemporal photostimulation patterns (Fig 7-8), and cellular resolution while simultaneously stimulating up to 50 targets in 3D (Fig 9). The supplement also includes a complete presentation of how to achieve proper control of power in individual targets (Fig 7). Additionally, we demonstrate 3D-SHOT in vivo in mice using electrophysiology (Fig 4). We hope that this additional data will convince the reviewer that 3D-SHOT is a powerful tool for biological applications.

However, the actual and the expected limits of the performance of the method is still not demonstrated in a convincing way but the authors seem rather to choose the optimal conditions for showing their results. I consider this misleading. I believe the reader would appreciate an unbiased representation of the limits at which the method (like any other method) will start to fail. This by no means would devalue the merits of their method but would rather provide a more rigorous and objective characterization. Here are some specific points:

1. Fig 8: The authors show the axial and lateral resolution in within a volume as a function of ROIs per hologram. What is more interesting is the lateral and spatial resolution as a function of depth (z). How is that for a population of neurons at 300um or 400um depth?

The authors show activation of 75 targets only. However, there must be more neurons within the shown volume than 75. Why can't they activate all neurons within that volume? And how would the spatial resolution look like if they activated 80 neurons that are not distributed far from each other in 3D in such a volume but rather within close proximity of each other, i.e. at the average distance of individual neurons in cortex?

To address these concerns we performed 3D measurements of two-photon induced fluorescence as previously shown in Figures 2 and 8, but parameterized holograms to show the radial and axial resolution of each spot in multi-ROI holograms as a function of 3D position (x,y,z), the number of spots, the mutual distance between spots, and propagation distance through scattering brain tissue (up to 800 um). We tested the system up to the point of failure to determine its operational limits.

By improving our hologram measurement systems we now present data on up to 750 spots and test the system up to its physical limits in terms of spatial range, number of targets, and optical aberrations (with mouse brain tissue slices). We used randomized, uniformly distributed point clouds to best replicate realistic experimental conditions. In addition, we have also refined our hologram measurement technique by placing the brain tissue slice directly in contact with the fluorescent monolayer, eliminating the coverslip between fluorescent layers and brain tissue, which distorted our previous measurements.

Using principal component analysis with repeated experiments, we identified the main factors driving spatial resolution and we quantified the operational limits of our experimental implementation of 3D-SHOT. Our findings are summarized below and have all been added to the text of the manuscript:

- 3D-SHOT preserves spatial resolution through up to 400-600 um of scattering brain slice tissue, which exceeds typical operational depths previously observed with similar two-photon imaging and photostimulation with temporal focusing.
- The absolute distance between targets does not significantly affect spatial resolution, and the system can operate at the native density of photosensitive neurons we observe with AAV viral expression. The

FWHM remains stable up until the point where the CTFP start to overlap when targets are selected that are closer than the dimensions of the CTFP.

- The target location has only a minor detectable effect on the FWHM. Using principal component analysis characterization with large populations of randomly distributed targets, we identified that the FWHM degrades as a function of target depth z , and we also identified minor effects between FWHM dimensions and location in the (x,y) plane. The effect of optical scattering being predominant, we also quantified position dependence of the FWHM independently of the presence of brain tissue.
- The system can distribute power on up to 380 targets in a single SLM phase mask without significant losses in spatial resolution. This estimated limit is not imposed by significant degradation of the average FWHM for two-photon response at the target locations. Rather, we observed that the target dimensions become more broadly distributed. Also, we detected a build up of unwanted diffuse illumination around non-targeted areas which relates to the overall quality of the point cloud hologram. In other words, the limiting factor imposing an upper bound on the number of simultaneous holographic targets is therefore given by a transition regime where losses in contrast may become the cause of undesired photocurrents in non-targeted areas. This limitation is specifically related to the resolution of the SLM, not 3D-SHOT *per se*, and this value would likely change when using an SLM with higher pixel density, and also with improved hologram computation methods. Also, this limitation is only imposed on each SLM phase mask, so the limit is 380 targets *per frame*. High frame rate SLMs would allow many ensembles of different neurons to be photo-stimulated sequentially.

These data are presented in the manuscript in Figure 8 and supplemental Figures S7, S8, and S9.

To further address the reviewer's question about how the PSFs would look through scattering medium, we have added a figure showing the physiological point spread function versus the depth below the pial surface for every neuron recorded *in vivo*. This figure shows that there is no relationship between cortical depth and resolution (within the volume tested, down to 325 μm) as predicted by volume imaging of two-photon induced fluorescence that indicates that scattering should only impact resolution between 400 and 600 μm . While the deepest neuron that we recorded from electrophysiologically was located 325 μm below the pial surface, this was limited only by the technical difficulty of targeted 2P patch deep in the brain, and not by the 3D-SHOT technique.

Across ROIs it seems that the axial FWHM is consistently lower (by about 10 μm) than the radial one. Why? And how can the authors justify that this is sufficient for activation of single neurons? I would expect that this would be even larger if neurons at 400 μm depth would be evaluated.

The axial FWHM is determined by the diffraction efficiency of the blazed grating used to induce temporal focusing by splitting femtosecond pulses into colors. As shown in Figure S2, there is a natural trade-off between hologram size or resolution, and temporal focusing. Although both can be simultaneously improved with a higher resolution SLM, here, we have adjusted the amount of temporal focusing for whole cell body photostimulation with single neuron spatial resolution, while maximizing the diffraction efficiency to preserve the overall quality of holograms. The amount of temporal focusing may be tuned differently for other applications if necessary.

The reason we chose the axial FWHM to be greater than the radial FWHM is that pyramidal neurons in L2/3 are larger in the axial dimension and extend apical dendrites along the optical axis that can contribute to the light induce photo-current. This effect is visible in figure 3, where the FWHM of photocurrent taken from flat CHO cells is narrower than the same hologram on neurons.

With regards to the reviewer's desire to evaluate neurons at 400 μm depth, we predict based on the data mentioned above that the PSF ought to start to degrade at around 400-500 μm . However, this is not a unique or interesting feature of 3D-SHOT, which is affected by scattering similarly to other temporal-focusing techniques. Indeed, the effect of optical aberrations through mouse brain tissue, and the advantages of temporal focusing for two-photon imaging and photostimulation through deep tissue have been well documented, [Papagiakoumou, et al. Nature photonics 7(4), 2013: 274-278.].

To further address the reviewer's question of how we can justify that this is sufficient for activation of single neurons, we have included a modeling experiment (Supplementary Figure S10) and amended the text (particularly in the discussion) to make clear that single-neuron spatial resolution is not something that is either achieved or not, but ought to be thought of on a continuum defined by the probability of eliciting off-target spikes given a particular patterned optogenetic stimulus. We make clear that 3D-SHOT does not *always* provide single-neuron resolution, but that it is capable of doing so in some contexts, and is an excellent strategy for single shot 3D patterned holographic stimulation while keeping off-target activation to a minimum.

Fig9: Again, it's not clear at what depth the results were obtained. If authors claim single neuron resolution of activation how is it justified that as shown in f) a FWHM of ~50um – not considering the depth at which the data was taken – still constitutes single neuron resolution? Also, here again only 50 targets were chosen, thus the same arguments and questions as above apply.

In figure 9F, the data are taken from brain slices, likely through ~50-100 um or less of scattering tissue. The reported FWHM of 50 um is measured using whole-cell recordings from photocurrent, and is therefore not as narrow as FWHM of light-evoked spiking which is sharpened by the additional nonlinearity of action-potential threshold. We show FWHM of photocurrent whenever possible because we believe it is a more faithful representation of the optical stimulus – quite the opposite from contriving to present our results under optimal conditions.

With regards to why 50 or 21 targets were chosen, as we explained in the previous rebuttal, we are limited by laser power available at the specimen. This has nothing to do with 3D-SHOT as a method *per se*, but is instead bounded by our laser power and our opsin's strength and light sensitivity. Measuring a fluorescent response from a slide requires much less laser power than activating a neuron *in vivo*; therefore we can assess the quality of up to 750 spots, but can only activate 21 neurons simultaneously. Increasing laser power and/or developing more sensitive microbial opsins will allow more targets to be simultaneously activated *in vivo*.

Thus, we hope the reviewer will understand that 50 holograms for photocurrent experiments and 21 holograms for spiking experiments are not arbitrary, and do not seek to present optimal conditions, but rather represent the maximum number of targets that we can activate physiologically given constraints that aren't directly related to how 3D-SHOT functions.

On a more general level, and this goes back to my comments in my previous report on the demonstration of the method in vivo, the authors need to convince the biological users that this is a method that can be used for actual biological studies and not a just a proof of principle. Thus, the authors need to show their method under actual conditions in which the method will be used, i.e. in vivo, at some reasonable depth, i.e. layer 2/3 neurons at least, for a sufficient large number of neurons and then show what are the conditions at which the method breaks down, i.e. at what depth, at what number spots, etc.

To further address this concern, we have added new experiments at the end of figure 9 showing axial and radial resolution *in vivo* in L2/3 while stimulating 21 neurons. Again, we only show 21 neurons because our available laser power and opsin sensitivity meant that our system could not *simultaneously* target more than 21 cells in a single shot exposure. Also, as previously mentioned, the depth of the neurons is now included in figure 9. These data conclusively show that 3D-SHOT can be used to simultaneously activate ensembles *in vivo* in L2/3 between 100-325 um. This ensemble stimulation in L2/3 is a reasonable use-case, as it is the use case in almost every other 2P optogenetics study.

If the intention of the authors is just to show proof of principles and the optical innovation then, I think a more specialized applied optics journal might be the better medium. In this context the authors seem to contradict themselves in their response: They say : "...have motivated the development of 3D-SHOT for specific neuroscience applications..." while at the same time the claim "Using 3D-SHOT in vivo to manipulate spatiotemporal neural activity is the subject of another manuscript which focuses on overcoming the biological challenges..."

We do not consider our statements as contradictory. We attempt to clarify our point:

A) *They say : "...have motivated the development of 3D-SHOT for specific neuroscience applications...*

This is true: we developed 3D-SHOT for optogenetics, not for imaging or other applications of temporal focusing. While it could be adapted for other uses, this is not the specific reason that we developed it. We had observed that temporal focusing was limited to 2D, that CGH provided poor axial resolution when designed for targeting large volumes, and that spiral scanning provided poor temporal resolution. We wanted to build a technology that had the spatial resolution of TF, the flexibility of CGH to target custom locations in 3D, along with a single shot approach that could simultaneously activate not only multiple neurons, but also the entire neuron soma to build strong photocurrents during a precisely timed exposure. The purpose of this manuscript is to demonstrate that these design specifications have been achieved, and that 3D-SHOT represents a practical technology for future applications in neuroscience research. We argue that our results represent a sufficient advance to warrant publication as it opens the door to many new types of neurobiological experiments not previously possible.

B) *Using 3D-SHOT in vivo to manipulate spatiotemporal neural activity is the subject of another manuscript which focuses on overcoming the biological challenges...*

This is also true. We present ensemble 3D stimulation *in vivo* using 3D-SHOT in this manuscript, improving the ability to control neural activity in space and time requires innovation in the properties of microbial opsins as well as the innovation in the optical approaches detailed in this manuscript. The second manuscript to which we refer is an opsins-engineering study where we adapt the properties of microbial opsins to be more suitable for 2P stimulation.

As stated in my previous report, to be convincing the authors need to show their method in action, i.e. to demonstrate the performance of their method and their claims in vivo in an unbiased fashion. Instead currently the authors pick individual aspects of their claim and show them under somewhat idealized conditions or while ignoring the effect on other parameters. Again, the method should be seen through the eyes of an end-user. There is a serious investment required to setup such a system in a systems neuroscience lab and ultimately the more convincing the authors can be to the end-users the higher will be the impact of their technique.

We have included additional data to address this point. In order to better show future users the capabilities of 3D-SHOT, we have conducted an additional set of experiments aimed at testing performance up to the point of failure.

Additionally, we have included new experiments, showing that 3D shot has high spatial resolution *in vivo* in layer 2/3 even when targeting multiple neurons. We now also report the depths of the neurons recorded *in vivo*.

Overall, this set of experiments, demonstrates that 3D-SHOT functions as presented and represents a major advance over currently available approaches. Not only we show how it can be put to use for practical photostimulation applications with existing opsins, but also, we have provided characterization experiments that describe performance independently of Opsin performance, to show the end-user what 3D-SHOT will be able to achieve when improved opsins will become available.

Reviewer #3 (Remarks to the Author):

We thank the reviewer for evaluating our manuscript. We have modified our manuscript to address the minor issues listed below.

The authors have addressed many of my concerns through changes to the manuscript, and additional clarification in their rebuttal letter, adequately. I was particularly excited about the characterization of the point-spread function of L2/3 neurons via 2P guided patch recordings *in vivo*.

Provided that some minor issues are corrected, as outlined below, I would feel more comfortable now recommending this manuscript for publication in Nature Communications.

Introduction:

1. The authors' claim in several locations in the manuscript (e.g. in the abstract) that their technique enables photoactivation 'in any number of focal planes'. In my understanding, the actual claim should be that the focal plane can be chosen from a continuum such that each stimulation location can lie in a unique focal plane if need be. It should be a statement about the unrestricted choice of stimulation planes (rather than about their unlimited total number).

We have updated the text to reflect that 3D-SHOT can stimulate any chosen stimulation planes, not an unbounded number.

2. The following claim in the intro is still overstated, please narrow it to reflect the actual advance this work is showing:

'Several methods have been developed for optogenetic photostimulation, however none are capable of simultaneous single neuron spatial resolution [...] while offering temporal precision across a large volume.'

Our goal is to emphasize that the development of 3D-SHOT was motivated by the need for a technology that satisfies all the specifications listed below:

- 1 - Single shot photostimulation (for precise timing of action potential triggering)
- 2 - Whole neuron soma illumination to yield strong photocurrents and take advantage of all available opsin.
- 3 - Custom targeting in 3D. Neurons of interest may be located anywhere within the operational range of the system.
- 4 - Optimal confinement to optimize spatial resolution down to the dimensions of individual neurons.
- 5 - Ability to operate through deep brain tissue

In the introduction, we described how existing technologies only partially satisfy at least one of these requirements, and we show how 3D-SHOT is a unique system design that is meant to succeed in all categories, by combining the advantages of temporal focusing for enhanced spatial confinement and for stability through brain tissue and by making it compatible with a simple form of Fourier Holography allowing custom targeting any desired locations, in 3D. We have therefore edited this sentence to emphasize that 3D-SHOT is designed to treat this set of equally critical constraints as a whole.

3. Modify/remove 'uses absorption in the visible spectrum to directly activate the opsin' from the description of 1P optogenetics. This is misleading as it implies that somehow 1P processes are direct and 2P processes are indirect, which is not a canonical use of terminology.

We have edited the text to fix this use of terminology.

4. 'Since two-photon absorption is a non-linear process proportional to the square of the light intensity' - not the process is proportional but its efficiency or probability etc.

We have edited the manuscript to address this concern.

5. I would write 'achieve/sustain photocurrent' rather than 'accumulate'

We have edited the manuscript to address this concern.

6. No need to write 'coherent laser beam' : laser beams are coherent by definition

We have edited the manuscript to address this concern.

7. In the explanation of temporal focusing 'depth of this layer' should be replaced by 'thickness of this layer'.

Selective photostimulation of vertically stacked neurons:

- The statement that the typical z-extent of a 'cortical pyramidal neuron is 25-30 um' is incorrect, at least as stated. Please specify species, layer, cortical area, and provide a reference.

We have edited the manuscript to address this concern, providing a reference for the soma size and instead of referencing a 25-30 um for the soma, we now reference that size for a soma in addition to a thick apical dendrite.

Simultaneously addressing arbitrary 3D locations...

-The authors use the term 'tomographic' to denote z-slices. Tomography typically refers to z-slices reconstructed from line-integrals via inverse Radon transform. This is not the case here, so please just say z-slice in lieu of 'tomographic image'.

We have edited the manuscript to address this concern.

Discussion:

- entire neuron's soma instead of 'entire neurons' soma'
- replace 'to spike neurons' by 'to activate neurons', 'elicit action potentials' or something similar.
- In (not 'with') CHO-cells [...] underlying photocurrent [...] linearly reflects the holographic patterns: Please rephrase this sentence to increase clarity. I understand that you wish to contrast the presence of just opsins in CHO vs the additional presence of voltage-sensitive channels in real neurons - just make this more clear & explicit.
- Even with a second SLM, the CTFP created will be identical for all stimulation locations at a given point in time. This should be stated more clearly at the end of the discussion.

We have edited the manuscript to address these concerns, providing additional clarity.

REVIEWERS' COMMENTS:

Reviewer #2 (Remarks to the Author):

In their revision, the authors have made significant improvements to their manuscript and have addressed my previous concerns thoroughly. I am particularly impressed by the new rigorous experiments (Fig 8, SI Fig. 7, 8 and 9) which show the regime within which 3D-SHOT can be used as a neuroscience tool. I think the community will find this manuscript a very useful technical resource and I do not have any further objections.

Reviewer #3 (Remarks to the Author):

The authors have adequately addressed my technical comments and suggestions, and I would like to commend them on the improvements made, and on the extra material included in the current submission. In my opinion, it should now be up to the community at large to further judge the scientific merit of this work. I am happy to recommend this manuscript for publication in its current form.

REVIEWERS' COMMENTS:

Reviewer #2 (Remarks to the Author):

In their revision, the authors have made significant improvements to their manuscript and have addressed my previous concerns thoroughly. I am particularly impressed by the new rigorous experiments (Fig 8, SI Fig. 7, 8 and 9) which show the regime within which 3D-SHOT can be used as a neuroscience tool. I think the community will find this manuscript a very useful technical resource and I do not have any further objections.

Reviewer #3 (Remarks to the Author):

The authors have adequately addressed my technical comments and suggestions, and I would like to commend them on the improvements made, and on the extra material included in the current submission. In my opinion, it should now be up to the community at large to further judge the scientific merit of this work. I am happy to recommend this manuscript for publication in its current form.

We thank the reviewers for their comments and feedback throughout the review process.